# Improved Algorithms for Nash Welfare in Linear Bandits

**Dhruv Sarkar** [1]  **Nishant Pandey** [2]  **Sayak Ray Chowdhury** [2]

## Abstract

Nash regret has recently emerged as a principled fairness-aware performance metric for stochastic multi-armed bandits, motivated by the Nash Social Welfare objective. Although this notion has been extended to linear bandits, existing results suffer from suboptimality in ambient dimension $d$, stemming from proof techniques that rely on restrictive concentration inequalities. In this work, we resolve this open problem by introducing new analytical tools that yield an order-optimal Nash regret bound in linear bandits. Beyond Nash regret, we initiate the study of $p$-means regret in linear bandits, a unifying framework that interpolates between fairness and utility objectives and strictly generalizes Nash regret. We propose a generic algorithmic framework, FairLinBandit, that works as a meta-algorithm on top of any linear bandit strategy. We instantiate this framework using two bandit algorithms: Phased Elimination and Upper Confidence Bound, and prove that both achieve sublinear $p$-means regret for the entire range of $p$. Extensive experiments on linear bandit instances generated from real-world datasets demonstrate that our methods consistently outperform the existing state-of-the-art baseline.

## 1. Introduction

In the classical stochastic multi-armed bandit (MAB) problem, the learner is typically evaluated by the arithmetic mean of expected rewards, or equivalently, by the standard notion of cumulative regret (Auer et al., 2002). While this objective is well-suited for maximizing total utility, it is inherently insensitive to how rewards are distributed across arms or agents. For instance, in a setting with two agents, a policy that allocates nearly all resources to one agent with slightly higher expected reward may achieve near-optimal arithmetic

regret, even if the other agent receives almost no reward. Such behavior is particularly problematic in high-stakes domains, such as clinical trials (Tewari & Murphy, 2017), where extreme disparities across patients are unacceptable, and fairness considerations are as critical as performance.

To address this limitation, a recent work has proposed *Nash regret* as a fairness-aware performance metric for stochastic bandit problems (Barman et al., 2023). Motivated by the Nash Social Welfare (NSW) objective – a metric well-known in the literature for satisfying key fairness axioms (Moulin, 2004) – Nash regret evaluates performance using the geometric mean of expected rewards rather than the conventional arithmetic mean. As a result, policies that severely disadvantage any agent incur large Nash regret, making the metric particularly effective at balancing efficiency and fairness.

The *p-means regret*, $p \in \mathbb{R}$, serves as a principled and flexible generalization of Nash regret (Krishna et al., 2025). The $p$-means welfare evaluates performance through the power mean of agents' expected rewards, thereby interpolating between different fairness–utility trade-offs as $p$ varies. In particular, smaller values of $p$ place greater emphasis on fairness by penalizing low-reward agents more strongly, while larger values of $p$ increasingly favor aggregate utility, recovering the standard arithmetic objective for $p = 1$. This unifying perspective subsumes Nash regret as a special case ($p = 0$) and enables a systematic study of fairness-aware learning objectives within a single analytical framework.

The Nash/$p$-means regret minimization problem has been studied for the stochastic $k$-armed bandit problem, and (almost) optimal regret bounds have been achieved (Barman et al., 2023; Krishna et al., 2025; Sarkar et al., 2025). However, the problem becomes substantially more challenging in the linear bandit setting since the number of arms is infinite and arms' rewards are dependent on each other. Here, each arm corresponds to a $d$-dimensional vector $x \in \mathbb{R}^d$, and its expected reward is linear in $x$, e.g., $\langle x, \theta^* \rangle$, for an unknown parameter vector $\theta^* \in \mathbb{R}^d$. Sawarni et al. (2023) were the first to analyze Nash regret for linear bandits. Their upper bound suffers from a strictly suboptimal dependence on the ambient dimension, scaling as $d^{5/4}$, whereas a known lower bound scales linearly with $d$ (Dani et al., 2008). This limitation is not merely a technical artifact of the analysis;

[1]Indian Institute of Technology Kharagpur [2]Indian Institute of Technology Kanpur. Correspondence to: Sayak Ray Chowdhury <sayakrc@iitk.ac.in>.

*Proceedings of the 43rd International Conference on Machine Learning*, Seoul, South Korea. PMLR 306, 2026. Copyright 2026 by the author(s).

rather, it stems from a fundamental bottleneck in algorithm design.

Consequently, the problem of attaining *order-optimal Nash regret* in linear bandits remained unresolved and was explicitly posed as an open question by Sawarni et al. (2023). At the same time, the broader problem of *minimizing p-means regret* in linear bandits had remained open. Against this backdrop, we make the following contributions:

(i) We resolve the open question of Sawarni et al. (2023) by introducing a meta-algorithm, FAIRLINBANDIT, to minimize Nash regret in linear bandits. We show that a carefully designed initial exploration phase with a data-adaptive stopping rule, followed by any optimal linear bandit algorithm for utility maximization, is sufficient to achieve *order-optimal* Nash regret $\widetilde{O}(d/\sqrt{T})$.

(ii) We achieve this result with new analytical techniques that overcome the previously unavoidable dimensional barriers. In particular, we depart from prior Nash regret analysis that requires reward-estimate dependent confidence intervals, necessitating restrictive multiplicative concentration inequalities (Sawarni et al., 2023). Instead, we work with UCB-style confidence intervals independent of estimated rewards, enabled by our novel data-adaptive stopping rule for the exploration phase.

(iii) Generalizing the techniques further, we show that FAIRLINBANDIT enjoys a $p-$means regret of $\widetilde{O}(d/\sqrt{T})$ for $p \geq 0$ and $\widetilde{O}(d/\sqrt{T} \cdot d^{|p|/2} \cdot \max\{1, |p|\})$ for $p < 0$. To the best of our knowledge, this is the first work to study $p$-means regret minimization in linear bandits, substantially broadening the scope and practical applicability of fairness-aware sequential learning algorithms.

(iv) Beyond yielding optimal dimension dependence, our approach broadens the applicability of welfare quantification in linear bandits to sub-Gaussian reward distributions, moving away from only non-negative (e.g., sub-Poisson) reward distributions considered in prior work (Sawarni et al., 2023). Moreover, our meta-algorithm FAIRLINBANDIT enables a generic reduction framework for Nash and $p$-means regret minimization with *any* optimistic linear bandit algorithm originally designed for average regret minimization.

(v) We instantiate FAIRLINBANDIT with two complementary algorithms: Linear Phased Elimination (LINPE) and Linear Upper Confidence Bound (LINUCB). We evaluate both variants on bandit instances derived from MSLR-WEB10K and Yahoo! Learning to Rank Challenge datasets. The results demonstrate that our algorithms consistently outperform the state-of-the-art LINNASH algorithm of Sawarni et al. (2023), achieving faster convergence and improved stability across a range of experimental settings.

A survey on related work is presented in Appendix, Sec. A.

## 2. Problem Statement

In the stochastic linear bandit problem, a learner interacts with the environment over $T$ rounds. The learner is given an action set $\mathcal{X} \subset \mathbb{R}^d$ which is compact and spans $\mathbb{R}^d$. Each arm $x \in \mathcal{X}$ is associated with a stochastic reward $r_x$ and its expected value $\mathbb{E}[r_x|x] = \langle x, \theta^* \rangle$ is a linear function of $x$ for some unknown parameter vector $\theta^* \in \mathbb{R}^d$. At each round $t$, the learner selects an arm $x_t \in \mathcal{X}$ and observes the corresponding reward $r_t := r_{x_t}$. Let $x^* \in \arg\max_{x \in \mathcal{X}} \langle x, \theta^* \rangle$ denotes an optimal arm with the highest expected reward. The learner's goal is to choose arms over a time horizon $T$ to minimize the *Nash regret*, defined as

$$\text{NR}_T := \max_{x \in \mathcal{X}} \langle x, \theta^* \rangle - \left( \prod_{t=1}^{T} \mathbb{E}[\langle x_t, \theta^* \rangle] \right)^{1/T}.$$

Unlike standard (average) regret, which is based on the arithmetic mean of expected rewards, Nash regret evaluates performance via the Nash Social Welfare (NSW) objective, i.e., the geometric mean of expected rewards. Defining the standard regret as $\text{AR}_T := \max_{x \in \mathcal{X}} \langle x, \theta \rangle - \frac{1}{T} \sum_{t=1}^{T} \mathbb{E}[\langle x_t, \theta^* \rangle]$, the AM–GM inequality implies that $\text{NR}_T \geq \text{AR}_T$ (Barman et al., 2023). Consequently, minimizing Nash regret is more challenging than minimizing average regret, while offering a stronger notion of fairness by discouraging policies that select arms with very low expected reward over the horizon.

The $p$-means regret, parameterized by $p \in \mathbb{R}$, generalizes Nash regret by evaluating performance via the power mean of expected rewards. It is defined as

$$\text{R}_T^p := \max_{x \in \mathcal{X}} \langle x, \theta \rangle - \left( \frac{1}{T} \sum_{t=1}^{T} \left( \mathbb{E}[\langle x_t, \theta \rangle] \right)^p \right)^{1/p}.$$

It captures a broad spectrum of fairness–utility trade-off, with the parameter $p$ maintaining the balance between these two conflicting objectives (Krishna et al., 2025).

For $p > 1$, the $p$-mean regret increasingly emphasizes rounds (e.g., individuals) with high rewards. In the limit $p \to \infty$, it becomes fully utilitarian, focusing solely on the best-off individual. At $p = 1$, it reduces to standard (arithmetic) regret, focusing on average utility over $T$ rounds. For $p < 1$, fairness considerations become more prominent: the power mean satisfies the Pigou–Dalton transfer principle (Moulin, 2004), favoring reward redistributions from better-off to worse-off rounds. In particular, $p$-means regret recovers Nash regret for $p = 0$, focusing on average welfare across $T$ rounds. As $p$ decreases further, it increasingly emphasizes rounds with low expected rewards. In the limit $p \to -\infty$, it becomes fully Rawlsian, focusing solely on the worst-off individual. Overall, smaller values of $p$ place greater emphasis on fairness, whereas larger values priori-

tize utility, with $p \leq 1$ being the primary regime of interest for fairness-aware learning objectives.

Throughout this work, we make the following assumptions.

**Assumption 1.** We assume that both the parameter vector and the arms are norm-bounded, i.e., $\|\theta^*\|_2 \leq 1$ and $\|x\|_2 \leq 1$ for all $x \in \mathcal{X}$. For each arm $x$, its observed reward $r_x$ is independent across realizations and is assumed to be $\sigma$-sub-Gaussian conditioned on $x$ with mean $\langle x, \theta^* \rangle$, namely,

$$\mathbb{E}\big[\exp\big(\zeta(r_x - \langle x, \theta^* \rangle)\big)\,|x\big] \leq \exp\big(\sigma^2 \zeta^2/2\big)\ , \quad \forall \zeta \in \mathbb{R}.$$

In addition, we assume that the expected rewards are non-negative, i.e., $\langle x, \theta^* \rangle \geq 0$ for all $x \in \mathcal{X}$.

The norm-boundedness and sub-Gaussian reward assumptions are standard in the linear bandit literature (Chu et al., 2011; Abbasi-Yadkori et al., 2011). The norm bounds can be relaxed to any other constants. The assumption of non-negative *expected* rewards $\mathbb{E}[r_x]$ is standard in the fairness-aware bandit literature (Barman et al., 2023; Krishna et al., 2025). It is necessary to define objectives, such as Nash regret and $p$-means regret, that rely on the geometric mean or the power mean. It also naturally arises in social welfare settings such as clinical trials or resource allocation, where arms are pre-screened to be, on average, beneficial.

*Remark* 2. Unlike prior work on fairness-aware linear bandits (Sawarni et al., 2023), we do *not* assume non-negative (random) realized rewards $r_x$. Allowing negative rewards (e.g., sub-Gaussian) is essential for modeling realistic outcomes; e.g., a treatment may be beneficial on average but can cause negative side effects in some instances.

**Notations.** Given a sample space $\Omega$, the probability simplex $\Delta(\Omega)$ on $\Omega$ and a discrete probability distribution $\lambda \in \Delta(\Omega)$, we define its support as $\mathrm{Supp}(\lambda) = \{x \in \Omega : \Pr_{X \sim \lambda}\{X = x\} > 0\}$. For a matrix $V \in \mathbb{R}^{d \times d}$ and a vector $a \in \mathbb{R}^d$, $\|a\|_V = \sqrt{a^\top V a}$ denotes the $V$-weighted norm.

## 3. Our Algorithm: FairLinBandit

In this section, we present FAIRLINBANDIT (Algorithm 1), a meta-algorithm for minimizing Nash and $p$-mean regret in linear bandits. The framework adopts a two-phase structure. The first phase focuses on sufficiently exploring the action set $\mathcal{X}$ while ensuring that expected rewards remain bounded away from zero, thereby preventing the geometric mean from collapsing. The second phase uses a regret-minimizing algorithm for linear bandits and exploits information acquired during Phase I while doing selective exploration.

### 3.1. Phase I (Exploration and Warm-up)

The goal of this phase is to obtain an accurate estimate of the unknown parameter $\theta^*$ while identifying a set of "good" arms and ensuring that the geometric mean of expected

rewards remains bounded away from zero. To achieve this, we interleave two complementary geometric strategies:

**(i) D-optimal design for information maximization.** To minimize uncertainty in estimating $\theta^*$, we find a distribution $\lambda^* \in \Delta(\mathcal{X})$ with support over at most $d(d+1)/2$ arms that maximizes $\log \det\big(U(\lambda)\big)$, where $U(\lambda) = \sum_{x \in \mathcal{X}} \lambda_x x x^\top$ denotes the design matrix. The objective is concave, allowing efficient computation of an optimal solution $\lambda^*$. To implement this design (known as the D-optimal design), we adopt a round-robin allocation for $\widetilde{T}$ rounds over the support of $\lambda^*$. Specifically, we pull each arm $z \in \mathrm{Supp}(\lambda^*)$ at least $\lceil \lambda_z^* \widetilde{T}/3 \rceil$ times, which ensures that the empirical design matrix $V$ satisfies $V \succeq \frac{\widetilde{T}}{3} U(\lambda^*)$. By Kiefer–Wolfowitz Theorem, $\lambda^*$ also minimizes the maximum predictive variance $g(\lambda) = \max_{x \in \mathcal{X}} \|x\|_{U(\lambda)^{-1}}^2$ (the G-optimal design objective), with $g(\lambda^*) = d$ (Lattimore & Szepesvári, 2020). As a consequence, for any arm $x \in \mathcal{X}$, the empirical predictive variance $x^\top V^{-1} x$ remains bounded by $3d/\widetilde{T}$.

**(ii) John ellipsoid-based robust initialization.** While arms pulled following D-optimal design guarantees uncertainty reduction, naively following it may lead to selecting arms with very small expected rewards, causing the geometric mean of rewards—and hence the Nash Social Welfare objective—to collapse. To prevent this, we incorporate a robust initialization strategy. Let $\mathrm{conv}(\mathcal{X})$ be the convex hull of the action set $\mathcal{X} \subset \mathbb{R}^d$ and $E$ the John ellipsoid of $\mathrm{conv}(\mathcal{X})$ with center $c$, satisfying $E \subseteq \mathrm{conv}(\mathcal{X}) \subseteq c + d(E - c)$. Via Caratheodory's theorem, the center $c$ can be expressed as a convex combination of $d + 1$ arms in $\mathcal{X}$, inducing a distribution $\rho \in \Delta(\mathcal{X})$ such that $\mathbb{E}_{x \sim \rho}[x] = c$ (Eckhoff, 1993). This guarantees an expected reward $\mathbb{E}[r_x]$ of at least $\langle x^*, \theta^* \rangle/(d+1)$, when $x \sim \rho$ (Sawarni et al., 2023).

At each round of Phase I, similar to Sawarni et al. (2023), our subroutine PULLARMS (Algorithm 2) flips a fair coin: with probability $1/2$, it selects an arm according to a D-optimal design (via round-robin allocation) to minimize estimation uncertainty, and with probability $1/2$, it samples an arm from the John ellipsoid distribution $\rho$ to preserve welfare. By interleaving the D-optimal design and the John ellipsoid distribution with equal probability, Phase I simultaneously achieves efficient exploration and welfare preservation.

The above interleaving strategy in Phase I, along with the so-called Nash confidence bound (e.g., a confidence bound that depends on reward estimates)-based optimistic algorithm in Phase II, however, doesn't guarantee an order-optimal Nash regret (Sawarni et al., 2023). While the authors conjecture that one would require fundamentally new analytical techniques that move beyond estimate-dependent confidence bounds, it was not immediately clear how to design such a technique that ensures the geometric mean of expected rewards doesn't collapse. We resolve this problem by intro-

**Algorithm 1** FAIRLINBANDIT (Meta-algorithm)

**Input:** Arm set $\mathcal{X}$, time horizon $T$, fairness parameter $p \in \mathbb{R}$, sub-Gaussian parameter $\sigma$.

1: Initialize $t = 1$, $V = [0]_{d \times d}$, $s = [0]_d$ and $\widehat{\theta} = [0]_d$.
2: // Phase I
3: Find the D-optimal design $\lambda^* \in \Delta(\mathcal{X})$ and John ellipsoid distribution $\rho \in \Delta(\mathcal{X})$ as described in Section 3.1, Phase I.
4: Set $p = 1$ if $p \geq -1$ and $\widetilde{T} = \lceil 72 \log T \rceil$.
5: **while** $t \leq \dfrac{48\sigma^2 d^2 \log T}{\left(\max\limits_{x \in \mathcal{X}} \langle x, \widehat{\theta} \rangle\right)^2}$ **or** $t \leq \dfrac{900p^2\sigma^2 d^2 \log T}{\left(\max\limits_{x \in \mathcal{X}} \langle x, \widehat{\theta} \rangle - \sqrt{\frac{48\sigma^2 d^2 \log T}{t}}\right)^2}$ **do**
6:     Update $(\widehat{\theta}, s, V) \leftarrow$ PULLARMS$(\rho, \lambda^*, \widetilde{T}, s, V)$
7:     Set $t \leftarrow t + \widetilde{T}$ and $\widetilde{T} \leftarrow 2\widetilde{T}$.
8: **end while**
9: // Phase II
10: Set $\tau = \widetilde{T}/2$ and run LINPE $(\tau, s, V)$ or LINUCB $(\tau, s, V)$ for the remaining $T - \tau$ rounds.

---

**Algorithm 2** PULLARMS (Subroutine in Phase I)

**Input:** Distributions $\rho, \lambda^*$, statistics $s, V$, sequence length $\widetilde{T}$

1: Initialize counts $c_z = 0$, $\forall z \in \mathcal{X}$
2: **for** $t = 1$ to $\widetilde{T}$ **do**
3:     With probability $1/2$ set flag = U, else, set flag = D.
4:     **if** flag = U or $\mathcal{A} = \emptyset$ **then**
5:         Sample $x_t$ randomly from the distribution $\rho$.
6:     **else if** flag = D **then**
7:         Pick the next arm $z$ in $\mathcal{X}$ (round robin) and set $x_t = z$.
8:         Set $c_z \leftarrow c_z + 1$.
9:         If $c_z \geq \lceil \lambda_z^* \widetilde{T}/3 \rceil$, then update $\mathcal{X} \leftarrow \mathcal{X} \setminus \{z\}$.
10:    **end if**
11:    Pull arm $x_t$, observe reward $r_t$
12:    Update $V \leftarrow V + x_t x_t^\mathsf{T}$, $s \leftarrow s + r_t x_t$.
13: **end for**
14: Compute estimate $\widehat{\theta} = V^{-1} s$
15: **return** $\widehat{\theta}, s, V$

---

**Algorithm 3** LINUCB (Subroutine in Phase II)

**Input:** Phase I length $\tau$, statistics $s, V$, regularizer $\alpha$.

1: Initialize $\overline{V}_\tau = V + \alpha I_d$ and $s_\tau = s$.
2: **for** $t = \tau + 1, \ldots, T$ **do**
3:     Compute regularized estimate $\widehat{\theta}_{t-1} = \overline{V}_{t-1}^{-1} s_{t-1}$.
4:     Pull arm $x_t = \arg\max_{x \in \mathcal{X}} \left\{ \langle x, \widehat{\theta}_{t-1} \rangle + \beta_{t-1} \|x\|_{\overline{V}_{t-1}^{-1}} \right\}$,
        where $\beta_{t-1} = \sigma\sqrt{d\log(1 + \frac{t-1}{d\alpha}) + 2\log T} + \sqrt{\alpha}$.
5:     Observe reward $r_t$.
6:     Update $\overline{V}_t = \overline{V}_{t-1} + x_t x_t^\mathsf{T}$ and $s_t = s_{t-1} + r_t x_t$.
7: **end for**

---

ducing a novel data-driven terminating condition for Phase I that facilitates the use of estimate-independent confidence bounds (e.g., the upper confidence bound) in Phase II.

**Data-adaptive terminating condition.** While the LIN-NASH algorithm of Sawarni et al. (2023) runs Phase I for a fixed, pre-decided number of rounds, we employ a *variable-length* exploration phase governed by a data-adaptive stopping rule. Specifically, Phase I of FAIRLINBANDIT continues as long as time index $t$ and the least-squares parameter estimate $\widehat{\theta}$ at time $t$ satisfy

$$t \leq \max\left( \frac{48\sigma^2 d^2 \log T}{\left(\max\limits_{x \in \mathcal{X}} \langle x, \widehat{\theta} \rangle\right)^2}, \frac{900p^2\sigma^2 d^2 \log T}{\left(\max\limits_{x \in \mathcal{X}} \langle x, \widehat{\theta} \rangle - \sqrt{\frac{48\sigma^2 d^2 \log T}{t}}\right)^2} \right) \quad (1)$$

At first glance, this adaptive stopping rule might appear incompatible with the round-robin allocation used to implement D-optimal design, which typically assumes prior knowledge of the exploration length $\widetilde{T}$. We resolve this tension via a doubling schedule. Phase I is initialized with $\lceil 72 \log T \rceil$ rounds and subsequently proceeds in epochs of geometrically increasing length. Importantly, the epochs are never restarted: all observations are retained, and the sufficient statistics $V$ and $s$ are continuously updated by accumulating $x_t x_t^\mathsf{T}$ and $r_t x_t$ at every round within the PULLARMS subroutine. The terminating condition (1) is checked only at the end of each epoch, with the time between each evaluation increasing geometrically.

This data-adaptive design ensures that Phase I terminates after $\tau = \widetilde{\Theta}\left(\frac{d^2}{\langle \theta^*, x^* \rangle^2}\right)$ rounds, which crucially helps us work with the upper confidence bound (UCB) interval in Phase II instead of the Nash confidence bound (NCB) interval. Since the UCB interval turns out to be tighter than the NCB interval, we obtain a tighter, order-optimal Nash regret.

The termination condition (1) is also central to bounding $p$-means regret for all $p \in \mathbb{R}$—with Nash regret as a special case—within a unified algorithmic framework. To this end, we normalize the fairness parameter to $p = 1$ whenever $p \geq -1$, since the Phase I stopping condition turns out to be independent of $p$ in this regime.

### 3.2. Phase II (Explore-exploit using Bandit Algorithms)

In this phase, the goal is to identify near-optimal arms and efficiently converge to the optimal action $x^*$. Our meta-algorithm FAIRLINBANDIT permits the use of any standard optimistic linear bandit algorithm for the remaining $(T - \tau)$ rounds in Phase II, initialized with the sufficient statistics $V$ and $s$ obtained from Phase I. In this work, we instantiate FAIRLINBANDIT using two representative algorithms: (a) Linear Upper Confidence Bound (LINUCB) and (b) Linear Phased Elimination (LINPE).

**(b) LINUCB:** This subroutine (Algorithm 3) follows the *optimism in the face of uncertainty* principle (Abbasi-Yadkori et al., 2011). At the start of Phase II (time $\tau$), LINUCB sets $s_\tau = s$ and $\overline{V}_\tau = V + \alpha I_d$. for some regularizer $\alpha > 0$. At each subsequent round, it selects the most optimistic arm $x_{t+1} = \arg\max_{x \in \mathcal{X}} \max_{\theta \in E_t} \langle x, \theta \rangle$ over the ellipsoid $E_t = \{\theta : \|\theta - \widehat{\theta}_t\|_{\overline{V}_t} \leq \beta_t\}$ with center $\widehat{\theta}_t = \overline{V}_t^{-1} s_t$ and ra-

dius $\beta_t$. The radius $\beta_t$ is set approximately as $O(\sigma\sqrt{d\log t})$ so that the unknown parameter $\theta^*$ lies in the ellipsoid $E_t$ for all rounds $t$ of Phase II with high probability. Consequently, for each of the expected rewards $\langle x, \theta^*\rangle$, we obtain the UCB $\langle x, \widehat{\theta}_t\rangle + \beta_t\|x\|_{\overline{V}_t^{-1}}$, which trades off exploitation with exploration effectively, where the arm selected at each round admits the highest UCB.

Intuitively, the extensive exploration in Phase I yields well-conditioned matrices $\overline{V}_t$ in Phase II, resulting in tight confidence ellipsoids. Consequently, the width $\beta_t\|x\|_{V_{t-1}^{-1}}$ of the UCB interval remains at most $\widetilde{O}(d/\sqrt{\tau})$ for all arms $x$, which eventually leads to order-optimal Nash regret.

*Remark* 3. The NCB intervals used by Sawarni et al. (2023) depend explicitly on estimated rewards, resulting in confidence widths that scale as $d^{5/4}$ in the dimension $d$ and yielding suboptimal Nash regret. The motivation for using NCB stems from the concern that UCB intervals may exceed the optimal reward $\langle x^*, \theta^*\rangle$, potentially causing the geometric mean to collapse by selecting arms with very small expected rewards. In contrast, our choice of Phase I length $\tau = \widetilde{\Theta}\left(\frac{d^2}{\langle \theta^*, x^*\rangle^2}\right)$ ensures that the UCB width $\widetilde{O}(d/\sqrt{\tau})$ does not exceed $\langle x^*, \theta^*\rangle$, eliminating this concern.

**(b) LINPE:** This subroutine (Algorithm 4) follows an episodic, doubling-window approach (Chu et al., 2011). After Phase I, we construct a surviving set of arms $\widetilde{\mathcal{X}}$ consisting of those that are sufficiently close to the current empirical leader $\gamma = \arg\max_{z\in\mathcal{X}}\langle z, \widehat{\theta}\rangle$. In particular, we retain only those arms satisfying $\langle x, \widehat{\theta}\rangle \geq \gamma - 8\sqrt{\frac{d^2\sigma^2\log T}{\tau}}$. Phase II then proceeds in episodes of geometrically increasing length using a doubling trick. In each episode, LINPE solves a new D-optimal design problem restricted to the surviving set $\widetilde{\mathcal{X}}$ and pulls each arm in the support of the resulting design for a number of rounds proportional to its weight $\lambda_a^*$. At the end of each episode, the estimate $\widehat{\theta}$ is updated using the newly observed rewards, the leader $\gamma$ is recomputed, and arms that no longer satisfy the confidence condition are eliminated.

Intuitively, once Phase I provides a sufficiently accurate estimate of $\theta^*$, Phase II focuses on efficiency: by restricting attention to $\widetilde{\mathcal{X}}$, every arm pulled is near-optimal. The doubling schedule allows the algorithm to progressively refine its estimates, ensuring an order-optimal Nash regret.

*Remark* 4. LINPE updates its parameter estimates only at the end of each episode, using them to eliminate suboptimal arms. In the next episode, it operates on the surviving set $\widetilde{\mathcal{X}}$, but discards all previously collected information. As a result, LINPE typically converges more slowly than LINUCB, which updates its estimates at every round while retaining the entire history. However, since LINPE restricts arm selection to the shrinking set of "good" arms $\widetilde{\mathcal{X}}$, it

---

**Algorithm 4** LINPE (Subroutine in Phase II)

**Input:** Phase I length $\tau$, statistics $s, V$.
1: Initialize $t = \tau + 1$, $T' = \frac{2}{3}\tau$ and $\widehat{\theta} = V^{-1}s$.
2: Find $\widetilde{\mathcal{X}} = \left\{x \in \mathcal{X} : \langle x, \widehat{\theta}\rangle \geq \max_{z\in\mathcal{X}}\langle z, \widehat{\theta}\rangle - 8\sqrt{\frac{d^2\sigma^2\log T}{\tau}}\right\}$.
3: **while** $t \leq T$ **do**
4:     Set $V \leftarrow [0]_{d\times d}$, $s \leftarrow [0]_d$ and find the D-optimal design $\lambda^* \in \Delta(\widetilde{\mathcal{X}})$ as described in Section 3.1.
5:     **for** all $a \in \mathsf{Supp}(\lambda)$ **do**
6:         Pull arm $a$ for the next $N_a = \lceil \lambda_a^* T'\rceil$ rounds.
7:         Observe $N_a$ rewards $z_1, \ldots, z_{N_a}$ and set $t \leftarrow t + N_a$.
8:         Update $V \leftarrow V + N_a \cdot aa^{\mathsf{T}}$ and $s \leftarrow s + (\sum_{j=1}^{N_a} z_j)a$.
9:     **end for**
10:    Compute $\widehat{\theta} = V^{-1}s$.
11:    Find $\widetilde{\mathcal{X}} = \left\{x \in \mathcal{X} : \langle x, \widehat{\theta}\rangle \geq \max_{z\in\mathcal{X}}\langle z, \widehat{\theta}\rangle - 8\sqrt{\frac{d^2\sigma^2\log T}{T'}}\right\}$.
12:    Update $T' \leftarrow 2T'$.
13: **end while**

---

enjoys significantly lower time complexity than LINUCB, which always selects from the full arm set $\mathcal{X}$. These trade-offs are confirmed by our experimental results.

## 4. Theoretical Results

The following theorem establishes an upper bound on the Nash regret of FAIRLINBANDIT. Both Phase II instantiations, LINUCB and LINPE, achieve the same order-wise bound (differing only in constants) since both rely on additive confidence widths, while the primary contribution to the regret bound is driven by Phase I.

**Theorem 5.** (Nash Regret of FAIRLINBANDIT) *Fix an action set $\mathcal{X} \subset \mathbb{R}^d, d \in \mathbb{N}$ and a moderately large time horizon $T$. Then, under Assumption 1, FAIRLINBANDIT, instantiated with either LINPE or LINUCB (with a regularizer $\alpha = O(1)$), enjoys a Nash regret*

$$\mathrm{NR}_T = O\left(\frac{\sigma d \log T}{\sqrt{T}}\right).$$

*Remark* 6 (Lower bound). The best-known lower bound for average regret in the linear bandit problem with infinitely many arms is $\Omega\left(d/\sqrt{T}\right)$ (Dani et al., 2008). By the AM–GM inequality, the Nash regret also admits the same lower bound. Our upper bound matches this lower bound up to polylogarithmic factors and is therefore order-optimal.

**Resolving an open problem.** The current state of the art is the LINNASH algorithm of Sawarni et al. (2023), whose Nash regret scales as $\widetilde{O}(d^{5/4}/\sqrt{T})$. As noted by the authors, this suboptimal dimension dependence stems from their use of specialized multiplicative concentration inequalities arising from confidence intervals (NCB) that depend on estimated rewards. In contrast, our data-adaptive termination rule (1) enables the construction of confidence intervals

that are independent of reward estimates, i.e., standard UCB-style intervals. This permits us to use self-normalized martingale concentration bounds (Abbasi-Yadkori et al., 2011) for LINUCB, or Hoeffding bounds for LINPE, which are tighter than multiplicative bounds. As a result, we obtain an improved, order-optimal dependence on the dimension $d$ and resolve the open problem of Sawarni et al. (2023).

Moreover, NCBs and multiplicative bounds restrict Sawarni et al. (2023) to work with non-negative rewards only. To do so, they consider sub-Poisson rewards rather than sub-Gaussian. While non-negative sub-Gaussian rewards are also sub-Poisson, we argue that their analysis breaks down even in this special case. See Appendix F for details.

**$p$-mean regret.** The next theorem establishes upper bounds on the $p$−mean regret of FAIRLINBANDIT for all $p \in \mathbb{R}$.

**Theorem 7.** *($p$−mean regret of FAIRLINBANDIT) Fix a fairness parameter $p \in \mathbb{R}$. Then, under the same premise of Theorem 5, the $p$-mean regret of* FAIRLINBANDIT *satisfies*

$$
R_T^p = \begin{cases} O\left( \frac{\sigma d \, \log T}{\sqrt{T}} \cdot d^{\frac{|p|}{2}} \cdot \max(1, |p|) \right), & p < 0 \,, \\ O\left( \frac{\sigma d \, \log T}{\sqrt{T}} \right), & p \geq 0 \,. \end{cases}
$$

To the best of our knowledge, this is the first result on $p$-mean regret minimization in linear bandits, substantially broadening the scope of our framework for studying per-round fairness in bandit algorithms.

**No free lunch.** The $p$-means regret matches the Nash regret rate $\widetilde{O}(d/\sqrt{T})$ for $p \geq 0$. Since the power mean is upper-bounded by the arithmetic mean for all $p \leq 1$, the $p$-means regret is lower-bounded by the average regret $\Omega(d/\sqrt{T})$ (Dani et al., 2008). Hence, our bound is order-optimal in the regime $p \in [0, 1]$. As $p$ decreases, corresponding to stricter fairness requirements, the regret bound degrades. For $p \in [-1, 0)$, the worst-case bound scales as $\widetilde{O}(d^{3/2}/\sqrt{T})$, while for $p < -1$, the regret grows exponentially with $|p|$. In particular, unless the horizon satisfies $T \geq \Omega(p^2 d^{|p|})$, the bound becomes vacuous as $p \to -\infty$. This behavior reveals an intrinsic fairness–performance trade-off: enforcing stronger fairness guarantees warrants a substantially longer time horizon, reflecting a "no free lunch" phenomenon. Our experimental results corroborate this theoretical prediction. Whether our regret upper bound is tight in the regime $p < -1$ remains an open question.

**Comparison with Sarkar et al. (2025).** The standard $k$-armed bandit problem with unknown means $\mu_1, \ldots, \mu_k$ can be viewed as a special case of $d$-dimensional linear bandits with $d = k$, where each arm corresponds to a canonical basis vector in $\mathbb{R}^k$ and the parameter vector is $\theta^* = (\mu_1, \ldots, \mu_k)$. In this setting, the best known bounds on $p$-means regret are $\widetilde{O}(\sqrt{k/T})$ for $p \geq 0$ and $\widetilde{O}(\sqrt{k/T} \cdot k^{|p|/2} \max\{1, |p|\})$ for $p < 0$ (Sarkar et al., 2025). Comparing these rates with

Theorem 7 reveals an additional $\sqrt{k}$ factor in our bounds. This gap arises because we operate in the more general setting of infinite arms, where arms' rewards are dependent on each other via a common parameter $\theta^*$, whereas Sarkar et al. (2025) considers a specialized setting of finitely many arms with mutually unrelated rewards.

### 4.1. A Reduction Framework

The impact of our data-driven termination rule for Phase I extends beyond achieving order-optimal Nash regret and resolving the associated open problem for infinite-armed linear bandits. More broadly, it yields a plug-and-play reduction framework for Nash regret minimization, allowing Phase II to be instantiated with *any* optimistic bandit algorithm designed for average regret minimization. All we need is: (i) an arm-independent bound on the confidence width $w_x := |\langle \theta^* - \widehat{\theta}, x \rangle|$ at the end of Phase I; (ii) a time-uniform upper bound on the per-round regret $\Delta_t := \langle \theta^*, x^* - x_t \rangle$ during Phase II; and (iii) a corresponding modification of the termination condition (1) to reflect these bounds.

For example, LINUCB satisfies $w_x \leq O\left(\sigma d \sqrt{\frac{\log \tau}{\tau}}\right)$ for all $x$ and $\Delta_t \leq O\left(\sigma d \sqrt{\frac{\log T}{\tau}}\right)$ for all $t > \tau$, yielding Phase I length $\tau = \Theta\left(\frac{\sigma^2 d^2 \log T}{\langle \theta^*, x^* \rangle^2}\right)$ and Nash regret $O\left(\frac{\sigma d \log T}{\sqrt{T}}\right)$. LINPE also exhibits similar bounds.

Instead, if we want to work with a randomized algorithm, e.g., Linear Thompson Sampling (LINTS), it can be plugged into Phase II (with a modified termination rule). It satisfies the same bound for confidence widths, but a weaker $\widetilde{O}\left(\frac{\sigma d^{3/2}}{\sqrt{\tau}}\right)$ bound for per-round regret, which would have resulted in a larger Phase I length $\widetilde{\Theta}\left(\frac{\sigma^2 d^3}{\langle \theta^*, x^* \rangle^2}\right)$ and a sub-optimal Nash regret $\widetilde{O}\left(\frac{\sigma d^{3/2}}{\sqrt{T}}\right)$. Note that LINTS is known to suffer from suboptimal average regret (Agrawal & Goyal, 2013), and hence, when plugged in our framework, also yields a suboptimal Nash regret.

If the number of arms is *finite*, say $k$, with the same linear reward structure, then we can plug in the SUPLINUCB algorithm of Chu et al. (2011) in Phase II, which is known to enjoy an order-optimal average regret $\widetilde{O}(\sigma \sqrt{d/T})$ in this setting. It satisfies $w_x \lesssim O\left(\sigma \sqrt{\frac{d \log(k\tau)}{\tau}}\right)$ for all $x$ and $\Delta_t \lesssim O\left(\sigma \sqrt{\frac{d \log(kT)}{\tau}}\right)$ for all $t > \tau$. This would lead to the stopping rule $t \leq \max\left( \frac{c_1 \sigma^2 d \log(kT)}{\left(\max_{x \in \mathcal{X}}\langle x, \widehat{\theta}\rangle\right)^2}, \frac{c_2 p^2 \sigma^2 d \log(kT)}{\left(\max_{x \in \mathcal{X}}\langle x, \widehat{\theta}\rangle - \sqrt{\frac{c_1 \sigma^2 d \log(kT)}{t}}\right)^2} \right)$ ($c_1, c_2$ are constants), which in turn would yield a Nash regret $O\left(\sigma \sqrt{\frac{d}{T}} \log(kT)\right)$, matching a known lower bound. We conjecture that this reduction framework could be extended to more general non-linear rewards (e.g., logistic).

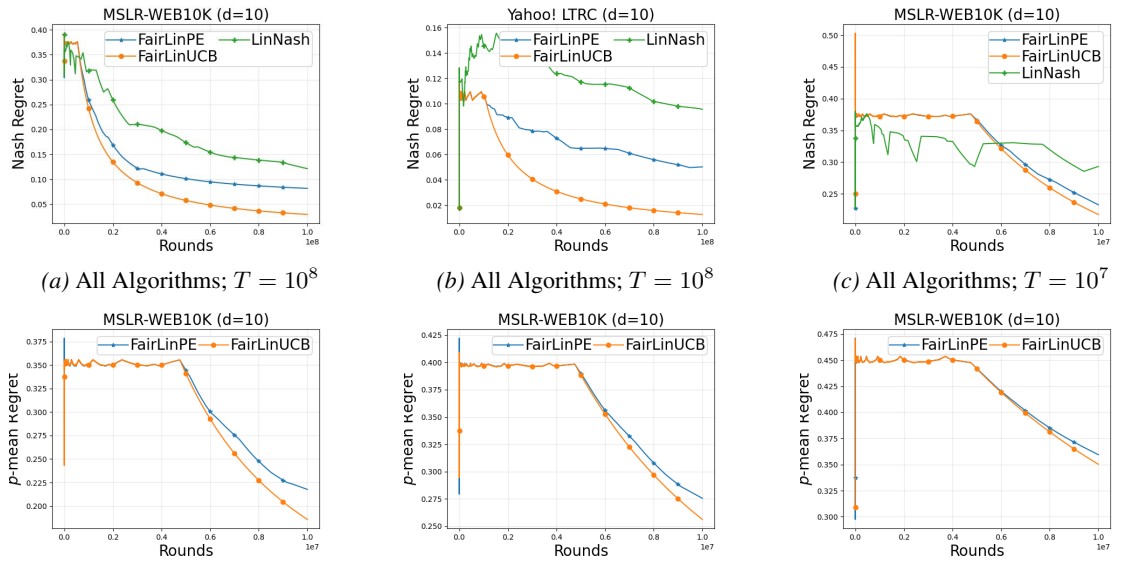

*(d)* FairLinPE vs FairLinUCB; $p = 0.5$ *(e)* FairLinPE vs FairLinUCB; $p = -0.5$ *(f)* FairLinPE vs FairLinUCB; $p = -1.5$

*Figure 1.* Numerical results comparing Nash regret for different algorithm runs. (a) and (b) showcase the better performance of FairLinPE and FairLinUCB over LinNash on MSLR-WEB10K and Yahoo! LTRC dataset, respectively for $T = 10^8$ rounds. (c) compares runs over $T = 10^7$ rounds and demonstrates the instability of LinNash for shorter time horizons. (d), (e), (f) compare the $p-$mean regret for FairLinPE and FairLinUCB at $p = 0.5$ , -0.5, and -1.5. The LinUCB procedure performs better than LinPE across all values of $p$.

## 5. Proof Techniques

We will provide a proof sketch for LINUCB (proof for LINPE is similar). We condition our analysis on the following good events that occur with high probability:

$\mathcal{G}_1$ : Phase I guarantees adequate exploration of the support of $D$-optimal design, ensuring that the estimated parameter $\widehat{\theta}$ remains within a tight confidence bound of the true parameter $\theta^*$ at all times.

$\mathcal{G}_2$ : Throughout Phase II under LINUCB, the unknown parameter $\theta^*$ lies within the confidence ellipsoids centered at the regularized least-squares estimates $\widehat{\theta}_t$ with radius $\beta_t$.

We refer the reader to Lemmas 14 and 23 in the Appendix for details of these good events. The next result bounds the length of Phase I (details in Lemma 17, Appendix).

**Lemma 8** (Informal; Number of Rounds in Phase I). *Under the event $\mathcal{G}_1$, Phase I runs for $\tau = \Theta\left(\frac{d^2}{\langle x^*,\theta^*\rangle^2}\right)$ rounds.*

We next ensure that if an arm is pulled in Phase II, then its mean must be close to the optimal mean $\langle x^*, \theta^*\rangle$, so that it does not significantly contribute to Nash or $p-$mean regret. See Lemma 24, 25, and 29 in the Appendix for details.

**Lemma 9** (Informal; Near optimality of Phase II arms). *If an arm $x_t$ is pulled in Phase II, then its instantaneous regret $\langle \theta^*, x^* - x_t\rangle$ is bounded by $O\left(\frac{d}{\sqrt{\tau}}\right)$ under the event $\mathcal{G}_2$.*

We now give short proof sketches of our regret bounds. See Appendix B.4 and C for detailed proofs.

*Proof Sketch (Theorem 5).* We split Nash Social Welfare as $\left(\prod_{t=1}^{\tau} \mathbb{E}[\langle x_t, \theta^*\rangle]\right)^{1/T}$ and $\left(\prod_{t=\tau+1}^{T} \mathbb{E}[\langle x_t, \theta^*\rangle]\right)^{1/T}$ corresponding to the two phases of the algorithm. The sampling strategy (mixing D-optimal design and John Ellipsoid distribution) ensures that for any $t \leq \tau$, $\mathbb{E}[\langle x_t, \theta^*\rangle] \geq \frac{\langle x^*,\theta^*\rangle}{2(d+1)}$ (Lemma 18 in the Appendix). Since Phase I runs for $\tau = \Theta\left(\frac{d^2}{\langle x^*,\theta^*\rangle^2}\right)$ rounds, the NSW in Phase I is lower bounded by $\langle x^*, \theta^*\rangle^{\frac{\tau}{T}} \left(1 - O(d^2/T)\right)$. In Phase II, we utilize the near-optimality guarantee. From the near-optimality lemma above, for any $t > \tau$, we have $\langle x_t, \theta^*\rangle \geq \langle x^*, \theta^*\rangle - \epsilon_t$, where $\epsilon_t \approx O(d/\sqrt{\tau})$. However, a tighter analysis (Lemma 26, 30 in the Appendix) shows that the cumulative effect over the remaining $T - \tau$ rounds ensures that the NSW in Phase II is lower bounded by $\langle x^*, \theta^*\rangle^{\frac{T-\tau}{T}} \left(1 - O\left(\frac{d\log T}{\sqrt{T}}\right)\right)$. Multiplying the bounds in both phases and substituting in the expression for Nash regret, we get $\mathrm{NR}_T \leq O\left(\frac{d\log T}{\sqrt{T}}\right)$. $\blacksquare$

*Proof Sketch (Theorem 7)* We analyze the cases for $p \geq 0$ and $p < 0$ separately. For $p \geq 0$, the result follows directly from the Generalized Mean Inequality, which establishes that the $p$-mean regret is upper bounded by the Nash regret ($p = 0$). For the regime $p < 0$ (setting $q = |p|$), we decompose the sum $\sum_{t=1}^{T}(\mathbb{E}[\langle x_t, \theta^*\rangle])^{-q}$ into contributions from Phase I and Phase II. In Phase I, the expected rewards are lower bounded by $\frac{\langle x^*,\theta^*\rangle}{2(d+1)}$, bounding the contribution $\sum_{t=1}^{T}(\mathbb{E}[\langle x_t, \theta^*\rangle])^{-q}$ from Phase I by approximately

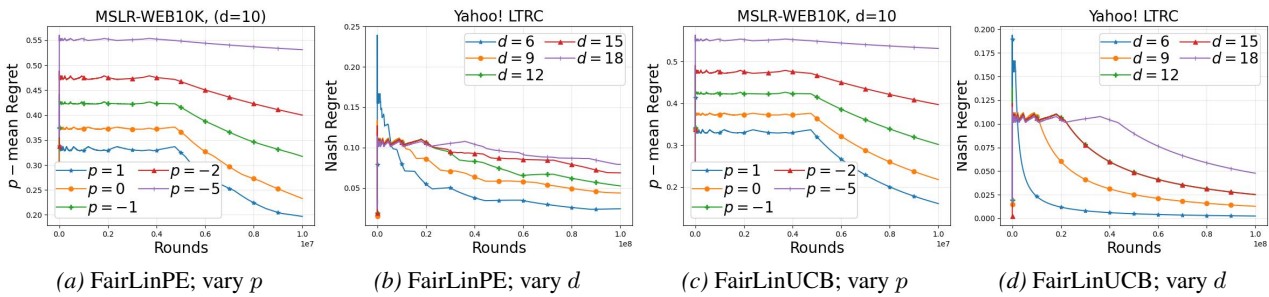

*Figure 2.* Numerical results showing the effect of variation of $p$ and $d$ on the regret. (a) and (c) show the effect of variation of $p$ on the $p-$ mean regret for FairLinPE and FairLinUCB, respectively, on the MSLR-WEB10K dataset. (b) and (d) show the effect of variation of $d$ on the Nash regret for FairLinPE and FairLinUCB, respectively, on the Yahoo! Learning To Rank Challenge dataset.

$\frac{\tau d^q}{T \langle x^*, \theta^* \rangle^q}$. In Phase II, we use the near-optimality condition $\langle x_t, \theta^* \rangle \geq \langle x^*, \theta^* \rangle - \epsilon_t$ and first-order approximations to linearize and upper bound the inverse rewards. The cumulative error term is then proportional to the sum of confidence widths, which is bounded by $\widetilde{O}(\sqrt{T})$ via the Cauchy-Schwarz inequality. Combining these components yields the final bound $O(d^{\frac{|p|+2}{2}} \log T / \sqrt{T})$.

# 6. Experiments

We evaluate the performance of our algorithms on bandit instances constructed from the MSLR-WEB10K (Qin & Liu, 2013) and Yahoo! Learning to Rank Challenge (Chapelle & Chang, 2011) datasets. We compare our proposed approaches, FAIRLINPE and FAIRLINUCB, against the LIN-NASH algorithm (Sawarni et al., 2023). All reported results are averages over 10 independent runs to estimate $\mathbb{E}[\langle x_t, \theta \rangle]$.

**Bandit instance construction.** We construct linear bandit instances by adapting the Learning-to-Rank (LTR) datasets into a non-contextual setting. The MSLR-WEB10K dataset contains 10,000 queries ($c$) and over 1.2 million relevance judgments, with up to 908 judged documents ($a$) per query. Each query-document pair $(c, a)$ is associated with a 135-dimensional feature vector $\phi(c, a)$. For each rank position $k \in \{1, \ldots, 908\}$, we compute the representative feature vector $a_k$ by averaging the features of all documents that appear at position $k$ across the entire set of queries. In this way, we obtain a set of 908 fixed arms with 135 features.

We simulate a linear bandit instance where the expected reward is governed by an unknown parameter $\theta^*$. We reduce

*Table 1.* Runtime for different algorithms on MSLR-WEB10K. FAIRLINPE runs faster than FAIRLINUCB and LINNASH.

| $T$ | LINNASH | FAIRLINPE | FAIRLINUCB |
|---|---|---|---|
| $10^6$ | 864.61s | 272.87s | 744.62s |
| $10^7$ | 1283.34s | 992.84s | 7225.71s |
| $10^8$ | 4213.14s | 3943.80s | 58201.23s |

the dimensionality of the arms to $d = 10$ using PCA and train a Lasso regression model on them to obtain $\theta^*$. We then normalize both $\theta^*$ and the arm vectors such that the mean rewards lie within the range $[0, 1]$. At each round $t$, the reward is sampled as $r_t = \langle x_t, \theta^* \rangle + \eta_t$, where $\eta_t$ is Gaussian with $\sigma = 0.5$. Note that LINNASH assumes sub-Poisson rewards (parameter $\nu$), we choose $\nu = 1$. We create another bandit instance following the same procedure for the Yahoo! LTRC dataset, which comprises 36,000 queries and 883,000 judgements, with up to 139 documents per query, with feature vectors of dimension 699.

**Comparison of Nash regret.** We first compare the Nash regret of our algorithms against LINNASH on both datasets with $T = 10^8$ and $d = 10$. As shown in Figure 1(a) (MSLR) and Figure 1(b) (Yahoo!), our algorithms minimize Nash regret significantly faster than LINNASH. To inspect performance at lower horizons, we plot Nash regret for $T = 10^7$ in Figure 1(c). The results indicate that LINNASH exhibits unstable performance in this regime. This instability arises because LINNASH relies on confidence intervals with a non-linear dependence on $d$; at lower horizons, these intervals remain too wide to eliminate suboptimal arms effectively. Please refer to Appendix E for a comprehensive comparision of the two algorithms with sub-Poission rewards.

**Comparison of $p$-mean regret.** We compare $p$-means regret of our two proposed algorithms in three distinct regimes: $p = 0.5$ ($0 < p \leq 1$), $p = -0.5$ ($-1 < p < 0$), and $p = -1.5$ ($p \leq -1$). Figures 1(d), 1(e), and 1(f) illustrate that FAIRLINUCB consistently outperforms FAIRLINPE, minimizing the $p$-mean regret faster across all three regimes.

**Ablation study.** We first analyze the impact of the fairness parameter $p$ on regret. Figures 2(a) and 2(c) show that, for both algorithms, the $p$-mean regret increases as $p$ decreases. This confirms the theoretical intuition that stricter fairness requirement (lower $p$) leads to higher regret. Next, using the Yahoo! dataset, we create bandit instances with dimensions $d \in \{6, 9, 12, 15, 18\}$ to study the effect of varying $d$ on Nash regret. Figures 2(b), 2(d) demonstrate a clear increase

in overall regret as the feature dimension increases, corroborating the linear dependence of Nash regret on $d$. Please refer to Appendix E for additional ablation studies.

**Computational costs.** We present a runtime (wall clock) comparison of different algorithms in Table 1, which highlights that FAIRLINPE is computationally faster than FAIRLINUCB. This, along with Figure 2, presents a clear trade-off: FAIRLINUCB offers better convergence, while FAIRLINPE offers superior computational efficiency. Moreover, we see that FAIRLINPE is faster than LINNASH and enjoys better performance. A rigorous study of computational complexities is presented in Appendix D.

## 7. Conclusion

Our work significantly advances the field of fairness-aware multi-armed bandits by resolving the open problem of suboptimal Nash regret bounds in the linear setting. By shifting from specialized multiplicative bounds to additive concentration inequalities, we achieve order-optimal Nash regret bounds and simultaneously pioneer the study of $p$-means regret to capture a broader spectrum of fairness-utility trade-offs. These theoretical contributions are operationalized through the FairLinBandit algorithmic framework, which is compatible with Phased Elimination (PE) and Upper Confidence Bound (UCB). Empirical evaluations on real-world datasets demonstrate that our proposed algorithms consistently outperform the state-of-the-art LinNash algorithm in terms of both speed and stability. An open question is to investigate the tightness of our regret bound in the regime $p < 0$ by characterizing lower bounds.

## Acknowledgements

SRC would like to thank Ayush Sawarni for initial discussions on this work and acknowledge the project ANRF/ARGM/2025/001456 from the Anusandhan National Research Foundation.

## Impact Statement

This paper presents work whose goal is to advance the field of machine learning. There are many potential societal consequences of our work, none of which we feel must be specifically highlighted here.

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

# Appendix

## A. Related Works

**Works on Social Welfare**   The notion of $p$-mean welfare is a well-established concept in the fair division literature, which lies at the intersection of mathematical economics and theoretical computer science. Rooted in social choice theory (Moulin, 2004), this welfare measure provides a flexible mechanism for balancing efficiency and equity, and has been studied extensively in prior work (Barman et al., 2020; Garg et al., 2021; Barman et al., 2022; Eckart et al., 2024). It is axiomatically characterized by five core properties—anonymity, scale invariance, continuity, monotonicity, and symmetry—which together ensure consistency with foundational fairness principles. In addition, the $p$-mean welfare satisfies the Pigou–Dalton transfer principle, whereby welfare strictly increases when resources are redistributed from better-off individuals to worse-off ones; this requirement restricts the parameter $p$ to be at most 1. Leveraging this strong theoretical foundation allows our approach to sidestep the introduction of arbitrary or ad-hoc fairness criteria.

**Works on Nash Regret**   Barman et al. (2023) initiated the study of Nash Regret in MAB. In the concluding note of their paper, Barman et al. (2023) mentioned that just like in Nash Regret geometric mean is used instead of arithmetic mean in the definition of regret, we can further generalize that by using generalized mean. Krishna et al. (2025) introduced the notion of p-means regret along the same lines and demonstrated regret bounds for different values of $p$. Unfortunately, the results in the paper were based on a problematic assumption, severely limiting the number of bandit instances to which the proposed algorithm was applicable. Sawarni et al. (2023) extended the idea of Nash Regret to Linear bandits and used important ideas from convex geometry to get meaningful regret guarantees in this setting. More recently, Sarkar et al. (2026) proposed differentially private algorithms that enjoy order-optimal Nash Regret bounds. Sarkar et al. (2025) demonstrated that specialized algorithm designs and strong assumptions, such as multiplicative concentration inequalities and strictly non-negative rewards, are not necessary to achieve near-optimal Nash regret in multi-armed bandits. They showed that a simple strategy consisting of an initial uniform exploration phase followed by a standard Upper Confidence Bound (UCB) algorithm—relying only on additive Hoeffding bounds—attains near-optimal Nash regret while naturally extending to sub-Gaussian reward distributions. This framework was further generalized to $p$-mean regret, providing nearly optimal bounds uniformly across all values of $p$.

## B. Nash Regret Analysis for FAIRLINBANDIT

### B.1. Phase I Analysis

**Lemma 10** (Chernoff Bound). *Let $Z_1, \ldots, Z_n$ be independent Bernoulli random variables. Consider the sum $S = \sum_{r=1}^{n} Z_r$ and let $\nu = \mathbb{E}[S]$ be its expected value. Then, for any $\varepsilon \in [0, 1]$, we have*

$$\Pr\{S \leq (1-\varepsilon)\nu\} \leq \exp\left(-\frac{\nu\varepsilon^2}{2}\right), \text{ and } \Pr\{S \geq (1+\varepsilon)\nu\} \leq \exp\left(-\frac{\nu\varepsilon^2}{3}\right).$$

**Lemma 11.** *During Phase* I, *at each time step $t \geq 72 \log T$, the arms from D-optimal design have been pulled at least $t/3$ times with probability at least $1 - \frac{1}{T}$.*

*Proof.* During Phase I, at each of the $\widetilde{T}$ rounds, an arm is selected independently according to the probability distribution $\lambda$ corresponding to the $D$-optimal design. Let $\mathcal{A}$ denote the support of $\lambda$.

For each round $i \in \{1, \ldots, \widetilde{T}\}$, define the indicator random variable

$$Z_i = \begin{cases} 1, & \text{if the arm selected at round } i \text{ belongs to } \mathcal{A}, \\ 0, & \text{otherwise.} \end{cases}$$

By construction, the random variables $\{Z_i\}_{i=1}^{\widetilde{T}}$ are independent Bernoulli random variables. Moreover, since the sampling distribution is $\lambda$ and $\mathcal{A}$ is its support, we have $\mathbb{E}[Z_i] = \sum_{a \in \mathcal{A}} \lambda(a) = 1/2$, where the last equality follows from the definition of Part I of the algorithm. Next, let $S_t = \sum_{i=1}^{t} Z_i$, which counts the total number of times an arm from $\mathcal{A}$ is

added to $S$ till time step $t$. The expected value of $S_t$ is therefore

$$\nu := \mathbb{E}[S_t] = \sum_{i=1}^{t} \mathbb{E}[Z_i] = t/2.$$

We wish to lower bound $S_t$. Observe that $\Pr\left(S_t \le \frac{t}{3}\right) = \Pr(S_t \le (1-\varepsilon)\nu)$, where $\varepsilon = 1/3$. Applying the lower-tail Chernoff bound from Lemma 11 yields

$$\Pr\{S_t \le (1-\varepsilon)\nu\} \le \exp\left(-\frac{\nu\varepsilon^2}{2}\right) = \exp\left(-\frac{t}{36}\right).$$

For any $t \ge 72 \log T$, we have $\exp\left(-\frac{t}{36}\right) \le \frac{1}{T^2}$. Hence $\Pr\left(S \ge \frac{t}{3}\right) \ge 1 - \frac{1}{T^2}$. Therefore, with probability at least $1 - \frac{1}{T^2}$, at a particular time step $t \ge 72 \log T$, the number of times arms from the $D$-optimal design have been pulled is $t/3$. Given that the number of time steps is at most $T$, applying the union bound, we get $1 - \frac{1}{T}$. $\qquad\square$

**Lemma 12** (Hoeffding Inequality). *Let $Z_1, \ldots, Z_n$ be independent random variables, with mean $\mu$ and subgaussianity parameter $\sigma$. Consider the empirical mean $\widehat{\mu} = \frac{1}{n} \sum_{r=1}^{n} Z_r$. Then, we have $\Pr\{|\widehat{\mu} - \mu| \ge \epsilon\} \le 2 \exp\left(-\frac{n\epsilon^2}{2\sigma^2}\right)$.*

**Lemma 13.** *Let $x_1, \ldots, x_t \in \mathbb{R}^d$ be a fixed set of vectors and let $r_1, \ldots, r_t$ be independent $\sigma$-sub-Gaussian random variables satisfying $\mathbb{E}[r_t] = \langle x_t, \theta^* \rangle$ for some $\theta^* \in \mathbb{R}^d$. Further, let $V_t = \sum_{j=1}^{t} x_j x_j^\mathsf{T}$ and $\widehat{\theta}_t = V_t^{-1}\left(\sum_j r_j x_j\right)$ be the least squares estimator of $\theta^*$ at round $t$. Then, for any $z \in \mathbb{R}^d$ and $\epsilon \ge 0$ we have*

$$\Pr\{|\langle z, \widehat{\theta}_t \rangle - \langle z, \theta^* \rangle| \ge \epsilon\} \le 2 \exp\left(\frac{-\epsilon^2}{2\sigma^2 z^\mathsf{T} V_t^{-1} z}\right).$$

*Proof.* Observe that

$$\widehat{\theta}_t - \theta^* = V_t^{-1} \sum_{j=1}^{t} r_j x_j - \theta^* = V_t^{-1} \sum_{j=1}^{t} r_j x_j - V_t^{-1} \sum_{j=1}^{t} \langle x_j, \theta^* \rangle x_j = V_t^{-1} \sum_{j=1}^{t} (r_j - \langle x_j, \theta^* \rangle) x_j.$$

$$\implies \langle z, \widehat{\theta}_t - \theta^* \rangle = z^\mathsf{T} V_t^{-1} \sum_{j=1}^{s} (r_j - \langle x_j, \theta^* \rangle) x_j = \sum_{j=1}^{t} (r_j - \langle x_j, \theta^* \rangle) z^\mathsf{T} V_t^{-1} x_j. \tag{2}$$

Now, observe that the quantity $r_j - \langle x_j, \theta^* \rangle = r_j - \mathbb{E}[r_j]$ is a zero mean $\sigma$ sub-gaussian random variable. Also, let $Q = \sum_{j=1}^{t} (r_j - \langle x_j, \theta^* \rangle) z^\mathsf{T} V_t^{-1} x_j$. Then, $Q$ is a weighted sum of i.i.d. zero-mean $\sigma$ sub-Gaussian random variables. Hence, via the extension of Lemma 12, we have

$$\Pr\left\{\left|\sum_{j=1}^{t} (r_j - \langle x_j, \theta^* \rangle) z^\mathsf{T} V_t^{-1} x_j\right| \ge \epsilon\right\} \le 2 \exp\left(\frac{-\epsilon^2}{2\sigma^2 \sum_{j=1}^{t} (z^\mathsf{T} V_t^{-1} x_j)^2}\right)$$

$$= 2 \exp\left(\frac{-\epsilon^2}{2\sigma^2 \sum_{j=1}^{t} (z^\mathsf{T} V_t^{-1} x_j x_j^\mathsf{T} (V_t^{-1})^\mathsf{T} z)}\right)$$

$$= 2 \exp\left(\frac{-\epsilon^2}{2\sigma^2 z^\mathsf{T} V_t^{-1} \left(\sum_{j=1}^{t} x_j x_j^\mathsf{T}\right) V_t^{-1} z}\right) = 2 \exp\left(\frac{-\epsilon^2}{2\sigma^2 z^\mathsf{T} V_t^{-1} z}\right)$$

Thus, via (2), we have $\Pr\left\{\left|\langle z, \widehat{\theta}_t - \theta^* \rangle\right| \ge \epsilon\right\} \le 2 \exp\left(\frac{-\epsilon^2}{2\sigma^2 z^\mathsf{T} V_t^{-1} z}\right)$. $\qquad\square$

**Lemma 14** (Good Event $\mathcal{G}_1$). *Let $x_1, \ldots, x_t$ be the set of arms pulled in Phase I of FAIRLINBANDIT up to timestep $t$ based on the subroutine PULLARMS and $r_1, \ldots, r_t$ be the corresponding rewards. Define $V_t = \sum_{j=1}^{t} x_j x_j^\mathsf{T}$, $s_t = \sum_{j=1}^{t} r_j x_j$ and $\widehat{\theta}_t = V_t^{-1} s_t$. Suppose Assumption 1 holds. Let $\mathcal{G}_1$ be the event that at each timestep $t \le \tau$ in Phase I, the arms from D-optimal design have been pulled at least $t/3$ times, and $\left\|\widehat{\theta}_t - \theta^*\right\|_{V_t} \le 4\sqrt{d\sigma^2 \log T}$. Then, we have $\Pr\{\mathcal{G}_1\} \ge 1 - \frac{3}{T}$.*

*Proof.* To prove the lemma, we first establish a bound on $\left\|\widehat{\theta}_t - \theta^*\right\|_{V_t}$ based on the arm vectors and the $V_t$ matrix. Since arm vectors lie in the unit ball in $\mathbb{R}^d$, we write $\left\|\widehat{\theta}_t - \theta^*\right\|_{V_t}$ in terms of this unit ball. Let $\mathcal{B}$ denote the unit ball in $\mathbb{R}^d$. Then

$$\left\|\widehat{\theta}_t - \theta^*\right\|_{V_t} = \left\|V_t^{\frac{1}{2}}(\widehat{\theta}_t - \theta^*)\right\|_2 = \max_{y \in \mathcal{B}} \langle y, V_t^{\frac{1}{2}}(\widehat{\theta}_t - \theta^*)\rangle.$$

We construct an $\varepsilon$-net for the unit ball, denoted as $\mathcal{C}_\varepsilon$. For any $y \in \mathcal{B}$, we define $y_\varepsilon(y) := \operatorname{argmin}_{b \in \mathcal{C}_\varepsilon} \|b - y\|_2$. We can now write

$$\left\|\widehat{\theta}_t - \theta^*\right\|_{V_t} = \max_{y \in \mathcal{B}} \langle y - y_\varepsilon, V_t^{\frac{1}{2}}(\widehat{\theta}_t - \theta^*)\rangle + \max_{y \in \mathcal{B}} \langle y_\varepsilon(y), V_t^{\frac{1}{2}}(\widehat{\theta}_t - \theta^*)\rangle$$

$$\leq \max_{y \in \mathcal{B}} \|y - y_\varepsilon\|_2 \left\|V_t^{\frac{1}{2}}(\widehat{\theta}_t - \theta^*)\right\|_2 + \max_{y \in \mathcal{B}} |\langle y_\varepsilon(y), V_t^{\frac{1}{2}}(\widehat{\theta}_t - \theta^*)\rangle|$$

$$\leq \varepsilon \left\|(\widehat{\theta}_t - \theta^*)\right\|_{V_t} + \max_{y_\varepsilon \in C_\varepsilon} |\langle y_\varepsilon, V_t^{\frac{1}{2}}(\widehat{\theta}_t - \theta^*)\rangle|.$$

We arrive at the last inequality by noting that $\max_{y \in \mathcal{B}} \|y - y_\varepsilon\|_2$ is less than $\varepsilon$, which follows from the properties of the $\varepsilon$-net. Further in the term $\max_{y \in \mathcal{B}} \langle y_\varepsilon(y), V_t^{\frac{1}{2}}(\widehat{\theta}_t - \theta^*)\rangle$ we supress the dependence of $y_\varepsilon$ on $y$ and write the equivalent formulation $\max_{y_\varepsilon \in C_\varepsilon} |\langle y_\varepsilon, V_t^{\frac{1}{2}}(\widehat{\theta}_t - \theta^*)\rangle|$. Rearranging, we obtain

$$\left\|\widehat{\theta}_t - \theta^*\right\|_{V_t} \leq \frac{1}{1 - \varepsilon} \max_{y_\varepsilon \in C_\varepsilon} |\langle V_t^{\frac{1}{2}} y_\varepsilon, \widehat{\theta}_t - \theta^*\rangle|. \tag{3}$$

Next, we show that $|\langle V_t^{\frac{1}{2}} y_\varepsilon, \widehat{\theta}_t - \theta^*\rangle|$ is small for all values of $y_\varepsilon$. This would naturally imply a bound on $\max_{y_\varepsilon \in C_\varepsilon} |\langle V_t^{\frac{1}{2}} y_\varepsilon, \widehat{\theta}_t - \theta^*\rangle|$. For this, consider any vector $z \in \mathbb{R}^d$ satisfying $\|z\|_2 \leq 1$. Using Lemma 13 with $z \to V_t^{\frac{1}{2}} z$ and $\epsilon = \sqrt{2\sigma^2 \log(T^2 |\mathcal{C}_\epsilon|) \|z\|^2}$, we have

$$\Pr\{|\langle V_t^{\frac{1}{2}} z, \theta^* - \widehat{\theta}_t\rangle| \geq \sqrt{2\sigma^2 \log(T^2 |\mathcal{C}_\epsilon|) \|z\|^2}\} \leq 2 \exp\left(-\frac{2\sigma^2 \log(T^2 |\mathcal{C}_\epsilon|) \|z\|^2}{2\sigma^2 z^\mathsf{T} V_t^{\frac{1}{2}} V_t^{-1} V_t^{\frac{1}{2}} z}\right) = \frac{2}{T^2 |\mathcal{C}_\epsilon|}$$

Thus, with probability atleast $1 - \frac{2}{T^2 |\mathcal{C}_\epsilon|}$, we have

$$|\langle V_t^{\frac{1}{2}} z, \theta^* - \widehat{\theta}_t\rangle| \leq \sqrt{2\sigma^2 \log(T |\mathcal{C}_\epsilon|) \|z\|^2} \leq \sqrt{2\sigma^2 \log(T^2 |\mathcal{C}_\epsilon|)} \qquad (\|z\|^2 \leq 1)$$

Using the above relation, we can guarantee the following for a particular $y_\epsilon$ with a probability $1 - \frac{2}{T^2 |C_\epsilon|}$,

$$\frac{1}{1 - \varepsilon} |\langle y_\varepsilon V_t^{\frac{1}{2}}, \widehat{\theta}_t - \theta^*\rangle| \leq \frac{1}{1 - \varepsilon} \sqrt{2\sigma^2 \log(T^2 |\mathcal{C}_\epsilon|)}$$

Taking a union bound over all elements in $\mathcal{C}_\varepsilon$ gives a probability bound of $1 - \frac{2}{T^2}$. Thus, we can guarantee

$$\frac{1}{1 - \varepsilon} \max_{y_\varepsilon \in C_\varepsilon} |\langle V_t^{\frac{1}{2}} y_\varepsilon, \widehat{\theta}_t - \theta^*\rangle| \leq \frac{1}{1 - \varepsilon} \sqrt{2\sigma^2 \log(T^2 |\mathcal{C}_\epsilon|)}$$

Next, we note that $|\mathcal{C}_\varepsilon| \leq \left(\frac{3}{\varepsilon}\right)^d$ (Lattimore & Szepesvári, 2020), and by choosing $\varepsilon = 1/2$ we get (for large enough $T$ and relatively large $d$)

$$\frac{1}{1 - \varepsilon} \max_{y_\varepsilon \in C_\varepsilon} |\langle V_t^{\frac{1}{2}} y_\varepsilon, \widehat{\theta}_t - \theta^*\rangle| \leq 4\sqrt{d\sigma^2 \log T}$$

Then we make use of inequality (3) to conclude $\left\|\widehat{\theta}_t - \theta^*\right\|_{V_t} \leq 4\sqrt{d\sigma^2 \log T}$. Finally, we take a union bound over all timesteps $\tau$ in Phase I. Since $\tau < T$, this, along with the event in Lemma 11 gives us $\Pr\{\mathcal{G}_1\} \geq 1 - \frac{2T}{T^2} - \frac{1}{T} = 1 - \frac{3}{T}$. $\qquad \square$

**Lemma 15** (Kiefer–Wolfowitz theorem (Lattimore & Szepesvári, 2020)). *Suppose $\mathcal{X}$ is compact and spans $\mathbb{R}^d$. Let $\Delta(\mathcal{X})$ be the set of probability distributions supported on $\mathcal{X}$. For any $\lambda \in \Delta(\mathcal{X})$, define $U(\lambda) = \sum_{x \in \mathcal{X}} \lambda_x x x^{\mathsf{T}}$. Then there exists a $\lambda^* \in \Delta(\mathcal{X})$ with support $\mathrm{Supp}(\lambda^*) \leq d(d+1)/2$ such that*

- *$\lambda^*$ is a maximizer of the D-optimal design objective $f(\lambda) = \log \det(U(\lambda))$.*

- *$\lambda^*$ is a minimizer of the G-optimal design objective $g(\lambda) = \max_{x \in \mathcal{X}} ||x||^2_{U(\lambda)^{-1}}$ and $g(\lambda^*) = d$.*

**Lemma 16.** *Suppose the hypothesis of Lemma 14 holds. Then, under the event $\mathcal{G}_1$, we have $|\langle x, \widehat{\theta}_t \rangle - \langle x, \theta^* \rangle| \leq 4\sqrt{\frac{3d^2\sigma^2 \log T}{t}}$ for all arms $x \in \mathcal{X}$ and for all rounds $t \leq \tau$ of Phase I.*

*Proof.* Fix some $t \leq \tau$. Note that $\langle x, \theta^* - \widehat{\theta}_t \rangle = \langle V_t^{-1/2} x, V_t^{1/2}(\theta^* - \widehat{\theta}) \rangle$. Then, using Hölder's inequality, we have

$$|\langle x, \theta^* - \widehat{\theta}_t \rangle| \leq ||x||_{V_t^{-1}} \left\|\theta^* - \widehat{\theta}_t\right\|_{V_t} . \tag{4}$$

Now, note that since $\mathcal{G}_1$ holds, Lemma 11 ensures that the number of D-optimal exploration rounds is $\geq t/3$. Algorithm 1 enforces a round-robin selection such that every arm $z$ in the support of $\lambda$ is pulled at least $\lceil \lambda_z t/3 \rceil$ times. By definition, $V_t$ is the sum of positive semi-definite matrices. We can lower bound $V_t$ by considering only the subset of rounds in which the algorithm explores using the D-optimal design. Note that the contribution to $V_t$ by each such arm $z$ in the D-optimal design is given as $\lceil \lambda_z \frac{t}{3} \rceil zz^{\mathsf{T}}$. Thus, we have:

$$V_t = \sum_{t=1}^{t} x_t x_t^{\mathsf{T}} \succeq \sum_{z \in \mathrm{Supp}(\lambda)} \left\lceil \lambda_z \frac{t}{3} \right\rceil zz^{\mathsf{T}} \qquad \text{(Ignoring non-D-optimal rounds)}$$

$$\succeq \sum_{z \in \mathrm{Supp}(\lambda)} \left( \lambda_z \frac{t}{3} \right) zz^{\mathsf{T}} = \frac{t}{3} \sum_{z \in \mathrm{Supp}(\lambda)} \lambda_z zz^{\mathsf{T}} = \frac{t}{3} U(\lambda).$$

Thus, we get $V_t \succeq \frac{t}{3} U(\lambda)$. This, along with Lemma 15, yields

$$x^{\mathsf{T}} V_t^{-1} x \leq x^{\mathsf{T}} \left( \frac{t}{3} U(\lambda) \right)^{-1} x = \frac{3}{t} ||x||^2_{U(\lambda)^{-1}} \leq \frac{3d}{t} \text{ for all } t \leq \tau . \tag{5}$$

Hence, we have $||x||_{V_t^{-1}} \leq \sqrt{\frac{3d}{t}}$, which along with Lemma 14, completes the proof. $\square$

**Lemma 17** (Number of Rounds in Phase I). *Let Phase I of* FAIRLINBANDIT *run for $\tau$ rounds and suppose the hypothesis of Lemma 14 holds. Then, under the event $\mathcal{G}_1$, we have $864\, dS \leq \tau \leq 3072\, dS$ almost surely, where $S := \frac{|p_a|^2 d\sigma^2 \log T}{(\langle x^*, \theta^* \rangle)^2}$, with $p_a = 1$ if $p \geq -1$ and $p_a = p$ if $p < -1$.*

*Proof.* Let $\tau = 864dS$. Then for all $t \leq \tau$, the arm $\widehat{x}_t = \arg\max_{x \in \mathcal{X}} \langle x, \widehat{\theta}_t \rangle$ satisfies

$$\langle \widehat{x}_t, \widehat{\theta}_t \rangle \leq \langle \widehat{x}_t, \theta^* \rangle + 4\sqrt{\frac{3d^2\sigma^2 \log T}{t}} \qquad \text{(via event } \mathcal{G}_1\text{)}$$

$$\implies \frac{t\langle \widehat{x}_t, \widehat{\theta}_t \rangle}{3d} \leq \frac{t\langle \widehat{x}_t, \theta^* \rangle}{3d} + \frac{4t}{3d}\sqrt{\frac{3d^2\sigma^2 \log T}{t}}$$

$$\leq \frac{\tau\langle \widehat{x}_t, \theta^* \rangle}{3d} + \frac{4}{3}\sqrt{3t\sigma^2 \log T}$$

$$= 288p_a^2 d\sigma^2 \frac{\langle \widehat{x}_t, \theta^* \rangle \log T}{(\langle x^*, \theta^* \rangle)^2} + \frac{4}{3}\sqrt{3t\sigma^2 \log T} \qquad (\tau = \tfrac{864|p_a|^2 d^2\sigma^2 \log T}{\langle x^*, \theta^* \rangle^2})$$

$$\leq 288p_a^2 d\sigma^2 \frac{\log T}{\langle x^*, \theta^* \rangle} + \frac{4}{3}\sqrt{3t\sigma^2 \log T} \qquad (\langle \widehat{x}_t, \theta^* \rangle \leq \langle x^*, \theta^* \rangle)$$

$$\leq 288p_a^2 d\sigma^2 \frac{\log T}{\langle \widehat{x}_t, \theta^* \rangle} + \frac{4}{3}\sqrt{3t\sigma^2 \log T} \qquad (\langle \widehat{x}_t, \theta^* \rangle \leq \langle x^*, \theta^* \rangle)$$

$$\leq 288p_a^2 d\sigma^2 \frac{\log T}{\langle \widehat{x}_t, \widehat{\theta}_t \rangle - 4\sqrt{\frac{3d^2\sigma^2 \log T}{t}}} + \frac{4}{3}\sqrt{3t\sigma^2 \log T}$$

$$\leq 300p_a^2 d\sigma^2 \frac{\log T}{\langle \widehat{x}_t, \widehat{\theta}_t \rangle - 4\sqrt{\frac{3d^2\sigma^2 \log T}{t}}} + \frac{4}{3}\sqrt{3t\sigma^2 \log T} \,.$$

Note that for the last inequality to hold, we must have $\langle \widehat{x}_t, \widehat{\theta}_t \rangle - 4\sqrt{\frac{3d^2\sigma^2 \log T}{t}} > 0$. This assumption is captured in our stopping condition. Hence, we have

$$\frac{t\max_{x \in \mathcal{X}}\langle x, \widehat{\theta}_t \rangle}{3d} \leq 300p_a^2 d\sigma^2 \frac{\log T}{\max_{x \in \mathcal{X}}\langle x, \widehat{\theta}_t \rangle - 4\sqrt{\frac{3d^2\sigma^2 \log T}{t}}} + \frac{4}{3}\sqrt{3t\sigma^2 \log T} \,.$$

Rearranging the last inequality, we get

$$\max_{x \in \mathcal{X}}\langle x, \widehat{\theta}_t \rangle - 4\sigma d\sqrt{\frac{3\log T}{t}} \leq \frac{900p^2\sigma^2 d^2 \frac{\log T}{t}}{\max_{x \in \mathcal{X}}\langle x, \widehat{\theta}_t \rangle - 4\sigma d\sqrt{\frac{3\log T}{t}}} \,. \tag{6}$$

Next, consider $\tau := 3072dS$. Then at $t = \tau$, we have the following

$$\langle \widehat{x}_t, \widehat{\theta}_t \rangle \geq \langle x^*, \widehat{\theta}_t \rangle \geq \langle x^*, \theta^* \rangle - 4\sqrt{\frac{3d^2\sigma^2 \log T}{\tau}} \qquad \text{(via event } \mathcal{G}_1\text{)}$$

$$= \langle x^*, \theta^* \rangle - \langle x^*, \theta^* \rangle\sqrt{\frac{48}{3072}} = \frac{7}{8}\langle x^*, \theta^* \rangle \,.$$

Hence, we have

$$\frac{\tau\langle \widehat{x}_t, \widehat{\theta}_t \rangle}{3d} \geq \frac{\tau\langle x^*, \widehat{\theta}_t \rangle}{3d} \geq \frac{7}{8}\langle x^*, \theta^* \rangle \cdot 1024S = \frac{896p_a^2 d\sigma^2 \log T}{\langle x^*, \theta^* \rangle} \,. \tag{7}$$

Now, consider the following terms:

$$320p_a^2 d\sigma^2 \frac{\log T}{\langle \widehat{x}_t, \widehat{\theta}_t \rangle - 4\sqrt{\frac{3d^2\sigma^2 \log T}{\tau}}} = 320p_a^2 d\sigma^2 \frac{\log T}{\frac{7}{8}\langle x^*, \theta^* \rangle - 4\sqrt{\frac{3d^2\sigma^2 \log T}{\tau}}}$$

$$\leq 320p_a^2 d\sigma^2 \frac{\log T}{\frac{6}{8}\langle x^*, \theta^* \rangle} = 430p_a^2 d\sigma^2 \frac{\log T}{\langle x^*, \theta^* \rangle} \tag{8}$$

and

$$\frac{4}{3}\sqrt{3\tau\sigma^2 \log T} = \frac{72|p_a|d\sigma^2 \log T}{\langle x^*, \theta^* \rangle} \leq \frac{72|p_a|^2 d\sigma^2 \log T}{\langle x^*, \theta^* \rangle} \,. \tag{9}$$

Adding inequality (8) and equality (9), we have

$$320p_a^2 d\sigma^2 \frac{\log T}{\langle \widehat{x}_t, \widehat{\theta}_t\rangle - 4\sqrt{\frac{3d^2\sigma^2 \log T}{\tau}}} + \frac{4}{3}\sqrt{3\tau\sigma^2 \log T}$$

$$= 430p_a^2 d\sigma^2 \frac{\log T}{\langle x^*, \theta^*\rangle} + \frac{72p_a^2 d\sigma^2 \log T}{\langle x^*, \theta^*\rangle} \leq \frac{896p_a^2 d\sigma^2 \log T}{\langle x^*, \theta^*\rangle} \leq \frac{\tau\langle \widehat{x}_t, \widehat{\theta}_t\rangle}{3d} \ . \qquad \text{(via inequality (7))}$$

Thus, from the above inequality, we have

$$\frac{\tau\langle \widehat{x}_t, \widehat{\theta}_t\rangle}{3d} \geq 320p_a^2 d\sigma^2 \frac{\log T}{\langle \widehat{x}_t, \widehat{\theta}_t\rangle - 4\sqrt{\frac{3d^2\sigma^2 \log T}{\tau}}} + \frac{4}{3}\sqrt{3\tau\sigma^2 \log T}$$

$$\implies \frac{\tau \max_{x\in\mathcal{X}}\langle x, \widehat{\theta}_t\rangle}{3d} \geq 300p_a^2 d\sigma^2 \frac{\log T}{\max_{x\in\mathcal{X}}\langle x, \widehat{\theta}_t\rangle - 4\sqrt{\frac{3d^2\sigma^2 \log T}{\tau}}} + \frac{4}{3}\sqrt{3\tau\sigma^2 \log T} \ . \qquad (10)$$

Rearranging the last inequality, we get

$$\max_{x\in\mathcal{X}}\langle x, \widehat{\theta}_t\rangle - 4\sigma d\sqrt{\frac{3\log T}{\tau}} \geq \frac{900p^2\sigma^2 d^2\frac{\log T}{\tau}}{\max_{x\in\mathcal{X}}\langle x, \widehat{\theta}_t\rangle - 4\sigma d\sqrt{\frac{3\log T}{\tau}}} \ . \qquad (11)$$

From inequalities (6) and (11), we observe that for $t \leq \tau = 864dS$, the arm $\widehat{x}_t = \arg\max_{x\in\mathcal{X}}\langle x, \widehat{\theta}_t\rangle$ satisfies $\max_{x\in\mathcal{X}}\langle x, \widehat{\theta}_t\rangle - 4\sigma d\sqrt{\frac{3\log T}{t}} \leq \frac{900p^2\sigma^2 d^2\frac{\log T}{t}}{\max_{x\in\mathcal{X}}\langle x, \widehat{\theta}_t\rangle - 4\sigma d\sqrt{\frac{3\log T}{t}}}$. Whereas, for $\tau \geq 3072dS$, the arm $\widehat{x}_t$ violates this condition. Thus, we conclude that the length of Phase I satisfies $864dS \leq \tau \leq 3072dS$. This completes the proof of the lemma. $\qquad\square$

**Lemma 18** (Sawarni et al. (2023)). *Let $c \in \mathbb{R}^d$ denote the center of the John ellipsoid [1] for the convex hull of $\mathcal{X}$. Let $\rho \in \Delta(\mathcal{X})$ be a probability distribution that satisfies $\mathbb{E}_{x\sim\rho}[x] = c$. Then, it holds that $\mathbb{E}_{x\sim\rho}[\langle x, \theta^*\rangle] \geq \frac{\langle x^*, \theta^*\rangle}{(d+1)}$, where $x^*$ is the optimal action for parameter $\theta^*$.*

**Fact 19** (Sarkar et al. (2025)). *For all real $x \in [0, 1/2]$ and $a \geq 0$, we have $(1-x)^a \geq (1-2ax)$.*

**Lemma 20** (Nash Welfare in Phase I). *Let Phase I of FAIRLINBANDIT run for $\tau$ rounds. Then the product of expected rewards in Phase I satisfies*

$$\left(\prod_{t=1}^{\tau} \mathbb{E}[\langle x_t, \theta^*\rangle]\right)^{\frac{1}{T}} \geq \langle x^*, \theta^*\rangle^{\frac{\tau}{T}}\left(1 - \frac{\log(2(d+1))\tau}{T}\right) \ .$$

*Proof.* Note that at each iteration of the subroutine PULLARMS, with probability 1/2, we pull an arm that is sampled according to $\rho$. From Lemma 18, we obtain that, for any round $t \leq \tau$, expected reward of the pulled arm $x_t$ must satisfy $\mathbb{E}[\langle x_t, \theta^*\rangle] \geq \frac{\langle x^*, \theta^*\rangle}{2(d+1)}$. Therefore, we get

$$\left(\prod_{t=1}^{\tau} \mathbb{E}[\langle x_t, \theta^*\rangle]\right)^{\frac{1}{T}} \geq \left(\frac{\langle x^*, \theta^*\rangle}{2(d+1)}\right)^{\frac{\tau}{T}} = \langle x^*, \theta^*\rangle^{\frac{\tau}{T}}\left(1 - \frac{1}{2}\right)^{\frac{\log(2(d+1))\tau}{T}} \geq \langle x^*, \theta^*\rangle^{\frac{\tau}{T}}\left(1 - \frac{\log(2(d+1))\tau}{T}\right),$$

where the last inequality holds due to Fact 19. $\qquad\square$

### B.2. Phase II Analysis for FAIRLINUCB

**Lemma 21** (Confidence Ellipsoid (Abbasi-Yadkori et al., 2011)). *Let $\{\mathcal{F}_t\}_{t=0}^{\infty}$ be a filtration. Let $\{\eta_t\}_{t=1}^{\infty}$ be a real-valued stochastic process such that $\eta_t$ is $\mathcal{F}_t$-measurable and conditionally $\sigma$-sub-Gaussian for some $\sigma > 0$, i.e., for all $\zeta \in \mathbb{R}$,*

---

[1]For a convex body $K \subset \mathbb{R}^d$, its John ellipsoid Grötschel et al. (2012) with center $c \in \mathbb{R}^d$ satisfies $E \subseteq K \subseteq c + d(E - c)$, where $c + d(E - c) = \{c + d(x - c) : x \in E\}$ denotes the dilation of $E$ by a factor of $d$.

$\mathbb{E}\left[e^{\zeta\eta_t} \mid \mathcal{F}_{t-1}\right] \leq \exp\left(\frac{\zeta^2\sigma^2}{2}\right)$. *Let $\{x_t\}_{t=1}^\infty$ be an $\mathbb{R}^d$-valued stochastic process such that $x_t$ is $\mathcal{F}_{t-1}$-measurable and $r_t = \langle x_t, \theta^*\rangle + \eta_t$. Define $\overline{V}_t = \alpha I_d + \sum_{s=1}^t x_s x_s^\top$ for some $\alpha > 0$, $s_t = \sum_{s=1}^t r_s x_s$ and $\widehat{\theta}_t = \overline{V}_t^{-1} s_t$. Assume $\|\theta^*\|_2 \leq 1$. Then, for any $\delta > 0$, with probability at least $1 - \delta$, the following holds for all $t \geq 1$:*

$$\|\widehat{\theta}_t - \theta^*\|_{\overline{V}_t} \leq \sigma\sqrt{\log\left(\frac{\det(\overline{V}_t)}{\det(\alpha I_d)}\right) + 2\log(1/\delta)} + \sqrt{\alpha}.$$

**Lemma 22** (Abbasi-Yadkori et al. (2011))**.** *Suppose $x_1, \ldots, x_t \in \mathbb{R}^d$ be such that $\|x_s\|_2 \leq 1$ for $1 \leq s \leq t$. Then*

$$\sum_{s=1}^t \|x_s\|_{\overline{V}_{s-1}^{-1}}^2 \leq 2\log\left(\frac{\det(\overline{V}_t)}{\det(\alpha I_d)}\right) \text{ and } \det(\overline{V}_t) \leq \left(\alpha + \frac{t}{d}\right)^d.$$

**Lemma 23** (Good Event $\mathcal{G}_2$)**.** *Let $x_1, \ldots, x_\tau$ be the set of arms pulled in Phase I of FAIRLINBANDIT based on the subroutine PULLARMS and $x_{\tau+1}, \ldots, x_t$ be the set of arms pulled up to time $t$ based on the subroutine LINUCB. Let $r_1, \ldots, r_t$ be the corresponding rewards. For some $\alpha > 0$, define $\overline{V}_t = \sum_{j=1}^t x_j x_j^\top + \alpha I_d$, $s_t = \sum_{j=1}^t r_j x_j$ and $\widehat{\theta}_t = \overline{V}_t^{-1} s_t$. Suppose Assumption 1 holds. Let $\mathcal{G}_2$ be the event that $\left\|\widehat{\theta}_t - \theta^*\right\|_{\overline{V}_t} \leq \beta_t$ for all rounds $t \in (\tau, T]$, where $\beta_t = \sigma\sqrt{d\log\left(1 + \frac{t}{d\alpha}\right) + 2\log T} + \sqrt{\alpha}$. Then, we have $\Pr\{\mathcal{G}_2\} \geq 1 - \frac{1}{T}$.*

*Proof.* First note that Lemma 22 gives $\frac{\det(\overline{V}_t)}{\det(\alpha I_d)} \leq \left(1 + \frac{t}{d\alpha}\right)^d$. Next observe that $x_t$ is $\mathcal{F}_{t-1}$-measurable and $\eta_t = r_t - \langle x_t, \theta^*\rangle$ is $\mathcal{F}_t$-measurable with respect to the $\sigma$-algebra $\mathcal{F}_t = \sigma(x_1, x_2, \ldots, x_{t+1}, \eta_1, \eta_2, \ldots, \eta_t)$. Moreover, from Assumption 1, $\eta_t$ is conditionally $\sigma$-sub-Gaussian. Hence, from Lemma 21, for any $\alpha > 0$ and $\delta \in (0, 1]$, we have

$$\forall t \in (\tau, T], \ \|\widehat{\theta}_t - \theta^*\|_{\overline{V}_t} \leq \sigma\sqrt{d\log\left(1 + \frac{t}{d\alpha}\right) + 2\log(1/\delta)} + \sqrt{\alpha}\,, \tag{12}$$

with probability at least $1 - \delta$. Setting $\delta = 1/T$, the proof concludes. $\square$

**Lemma 24** (Instantaneous Regret Bound)**.** *Suppose Assumption 1 holds. Then, under the event $\mathcal{G}_2$, the instantaneous regret $\Delta_t = \langle\theta^*, x^*\rangle - \langle\theta^*, x_t\rangle$ satisfies $\Delta_t \leq 2\beta_{t-1}\min\left(\|x_t\|_{\overline{V}_{t-1}^{-1}}, 1\right)$ for all $t \in (\tau, T]$.*

*Proof.* By Hölder's inequality and from Lemma 23, under the evnet $\mathcal{G}_2$, we have

$$\left|\langle\widehat{\theta}_{t-1} - \theta^*, x\rangle\right| \leq \|\widehat{\theta}_{t-1} - \theta^*\|_{\overline{V}_{t-1}}\|x\|_{\overline{V}_{t-1}^{-1}} \leq \beta_{t-1}\|x\|_{\overline{V}_{t-1}^{-1}}\,.$$

This, along with the UCB arm selection rule, yields the instantaneous regret

$$\Delta_t = \langle\theta^*, x^*\rangle - \langle\theta^*, x_t\rangle \leq \langle\widehat{\theta}_{t-1}, x^*\rangle + \beta_{t-1}\|x^*\|_{\overline{V}_{t-1}^{-1}} - \langle\theta^*, x_t\rangle$$

$$\leq \langle\widehat{\theta}_{t-1}, x_t\rangle + \beta_t\|x_t\|_{\overline{V}_{t-1}^{-1}} - \langle\theta^*, x_t\rangle \leq 2\beta_{t-1}\|x_t\|_{\overline{V}_{t-1}^{-1}}\,.$$

Since $\Delta_t \leq 2$ from Assumption 1, we obtain $\Delta_t \leq 2\min\left(\beta_{t-1}\|x_t\|_{\overline{V}_{t-1}^{-1}}, 1\right) \leq 2\beta_{t-1}\min\left(\|x_t\|_{\overline{V}_{t-1}^{-1}}, 1\right)$. $\square$

**Lemma 25.** *Under the event $\mathcal{G}_1$, we have $\|x\|_{\overline{V}_t^{-1}} \leq \sqrt{\frac{3d}{\tau}}$ for all arms $x \in \mathbb{R}^d$ and for all rounds $t \in (\tau, T]$.*

*Proof.* Let $V_\tau = \sum_{j=1}^\tau x_j x_j^\top$ denote the design matrix at the end of Phase I. Note that for any $t \in [\tau, T]$, we have $\overline{V}_t \succeq \overline{V}_\tau \succeq V_\tau$. Thus, under the event $\mathcal{G}_1$, we obtain from (5), that $x^\top \overline{V}_t^{-1} x \leq x^\top V_\tau^{-1} x \leq \frac{3d}{\tau}$, and hence $\|x\|_{\overline{V}_t^{-1}} \leq \sqrt{\frac{3d}{\tau}}$ for any $x \in \mathbb{R}^d$. $\square$

**Lemma 26** (Nash Welfare in LINUCB). *Let $\mathcal{G} = \mathcal{G}_1 \cap \mathcal{G}_2$. Suppose Assumption 1 holds. Then, under the event $\mathcal{G}$, the product of expected reward in* LINUCB *satisfies*

$$\left( \prod_{t=\tau+1}^{T} \mathbb{E}[\langle x_t, \theta^* \rangle] \right)^{\frac{1}{T}} \geq \langle x^*, \theta^* \rangle^{\frac{T-\tau}{T}} \left( 1 - 16 \sqrt{\frac{d^2 \sigma^2}{T \langle x^*, \theta^* \rangle^2} \log T} \right)$$

*Proof.* First, from Lemma 24, we have under the event $\mathcal{G}_2$,

$$\langle x_t, \theta^* \rangle = \langle x_t, \theta^* \rangle - \Delta_t \geq \langle x^*, \theta^* \rangle - 2\beta_{t-1} \min\left( \|x_t\|_{\overline{V}_{t-1}^{-1}}, 1 \right) \geq \langle x^*, \theta^* \rangle - 2\beta_{t-1} \|x_t\|_{\overline{V}_{t-1}^{-1}}.$$

Therefore, we have

$$\left( \prod_{t=\tau+1}^{T} \mathbb{E}[\langle x_t, \theta^* \rangle] \right)^{\frac{1}{T}} \geq \prod_{t=\tau+1}^{T} \left( \langle x^*, \theta^* \rangle - 2\beta_{t-1} \mathbb{E}\left[ \|x_t\|_{\overline{V}_{t-1}^{-1}} \right] \right)^{\frac{1}{T}}$$

$$\geq \langle x^*, \theta^* \rangle^{\frac{T-\tau}{T}} \prod_{t=\tau+1}^{T} \left( 1 - \frac{2\beta_T \mathbb{E}\left[ \|x_t\|_{\overline{V}_{t-1}^{-1}} \right]}{\langle x^*, \theta^* \rangle} \right)^{\frac{1}{T}},$$

where the last inequality holds because $(\beta_t)_{t>\tau}$ is an increasing sequence. Now, from Lemma 25, under the event $\mathcal{G}_1$, we have $\|x_t\|_{\overline{V}_t^{-1}} \leq \sqrt{\frac{3d}{\tau}}$. Furthermore, for large enough $T$ and $\alpha = O(1)$, we get $\beta_T \leq \sqrt{4d\sigma^2 \log T}$. This gives $\frac{2\beta_T \|x_t\|_{\overline{V}_{t-1}^{-1}}}{\langle x^*, \theta^* \rangle} \leq \frac{\sqrt{48 d^2 \sigma^2 \log T}}{\sqrt{\tau} \langle x^*, \theta^* \rangle}$. From Lemma 17, under the event $\mathcal{G}_1$, we have $\tau \geq \frac{864 d^2 \sigma^2 \log T}{(\langle x^*, \theta^* \rangle)^2}$ since by our design $p_a = 1$ when $p = 0$. This yields $\frac{2\beta_T \mathbb{E}[\|x_t\|_{\overline{V}_{t-1}^{-1}}]}{\langle x^*, \theta^* \rangle} \leq 1/2$ and thus, invoking Fact 19, we get

$$\left( \prod_{t=\tau+1}^{T} \mathbb{E}[\langle x_t, \theta^* \rangle] \right)^{\frac{1}{T}} \geq \langle x^*, \theta^* \rangle^{\frac{T-\tau}{T}} \prod_{t=\tau+1}^{T} \left( 1 - \frac{4\beta_T \mathbb{E}\left[ \|x_t\|_{\overline{V}_{t-1}^{-1}} \right]}{T \langle x^*, \theta^* \rangle} \right)$$

$$\geq \langle x^*, \theta^* \rangle^{\frac{T-\tau}{T}} \left( 1 - \frac{4\beta_T \mathbb{E}\left[ \sum_{t=\tau+1}^{T} \|x_t\|_{\overline{V}_{t-1}^{-1}} \right]}{T \langle x^*, \theta^* \rangle} \right),$$

where the last inequality follows by using $(1-x)(1-y) \geq (1-x-y) \; \forall \; x, y > 0$. Now, from the Cauchy-Schwartz inequality and from Lemma 22, we have

$$\sum_{t=\tau+1}^{T} \|x_t\|_{\overline{V}_{t-1}^{-1}} \leq \sqrt{(T-\tau) \sum_{t=\tau+1}^{T} \|x_t\|_{\overline{V}_{t-1}^{-1}}^2} \leq \sqrt{2T \log\left( \frac{\det(\overline{V}_T)}{\det(\alpha I_d)} \right)} \leq \sqrt{2Td \log\left( 1 + \frac{T}{d\,\alpha} \right)} \leq \sqrt{4Td \log T},$$

for a large enough $T$ and $\alpha = O(1)$. Now we can further simplify the obtained expression as

$$\left( \prod_{t=\tau+1}^{T} \mathbb{E}[\langle x_t, \theta^* \rangle] \right)^{\frac{1}{T}} \geq \langle x^*, \theta^* \rangle^{\frac{T-\tau}{T}} \left( 1 - 16 \sqrt{\frac{d^2 \sigma^2}{T \langle x^*, \theta^* \rangle^2} \log T} \right).$$

The proof concludes by noting that the above holds under the event $\mathcal{G}$. $\qquad \square$

### B.3. Phase II Analysis for FAIRLINPE

**Lemma 27** (Good Event $\mathcal{G}_2$). *Let $x_1, \ldots, x_t$ be the set of arms pulled in episode $\ell$ of the subroutine* LINPE *and $r_1, \ldots, r_t$ be the corresponding rewards, where $t = \frac{2^\ell \tau}{3}$. Define $V_\ell = \sum_{j=1}^{t} x_j x_j^\mathsf{T}$, $s_\ell = \sum_{j=1}^{t} r_j x_j$ and $\widehat{\theta}_\ell = V_\ell^{-1} s_\ell$. Suppose Assumption 1 holds. Let $\mathcal{G}'_2$ be the event that $\left\| \widehat{\theta}_\ell - \theta^* \right\|_{V_\ell} \leq 4\sqrt{d\sigma^2 \log T}$ for all episodes $\ell$. Then, $\Pr\{\mathcal{G}'_2\} \geq 1 - \frac{2}{T}$. $|\langle x, \widehat{\theta}_\ell \rangle - \langle x, \theta^* \rangle| \leq 4\sqrt{\frac{3d^2 \sigma^2 \log T}{2^\ell \tau}}$ for all arm $x$ in the surviving arm set $\widetilde{\mathcal{X}}$ and for all episodes $\ell$.*

*Proof.* The proof is similar to that of Lemma 14. The reason the proof for this lemma (Phase II) is nearly identical to Lemma 14 (Phase I) is that both lemmas address the same fundamental mathematical challenge: bounding the deviation of a Least-Squares Estimator ($\hat{\theta}$) from the true parameter ($\theta^*$) in a linear setting. The discretization argument, the geometry of the arm space ($\mathbb{R}^d$), and the concentration properties of the noise remain constant regardless of which phase the algorithm is in. This is especially true because we are analyzing sampling via D-optimal design in both parts.

We begin with the high probability concentration bound of the estimate $\hat{\theta}$ computed at the end of every episode and then take a union bound over the number of episodes. A very loose upper bound on the number of episodes is $T$ and thus we get a probability bound of $1 - \frac{2}{T}$ on $\mathcal{G}_2'$.

Similarly, by the same analogy as Lemma 16, for every episode in Phase II with $T' = 2^\ell \tau / 3$ we have $\|x\|_{V_\ell^{-1}} \leq \sqrt{\frac{d}{T'}}$. The reason why a factor of 3 does not appear in the numerator of the fraction in the square root is that we know deterministically that the number of times arms from the $D$-optimal design are samples is $T'$. However, in Lemma 16 we had to resort to a high probability lower bound of a third of the number of timesteps upto that point.

Finally, using bounds on $\left\|\theta^* - \widehat{\theta}_\ell\right\|_{V_\ell}$ from event $\mathcal{G}_2'$, and substituting in (4), we get the desired bound. $\qquad\square$

**Lemma 28.** *Suppose the hypothesis of Lemma 27 holds. Then, under the event $\mathcal{G}_2'$, the optimal arm $x^*$ always exists in the surviving set $\widetilde{\mathcal{X}}$ in every episode $\ell$.*

*Proof.* Let $t = T' = 2^\ell \tau / 3$ for every episode $\ell$ in Phase II. From Lemma 27 we have

$$\langle x^*, \widehat{\theta}_t \rangle \geq \langle x^*, \theta^* \rangle - 4\sqrt{\frac{d^2 \sigma^2 \log T}{t}} \geq \langle x, \theta^* \rangle - 4\sqrt{\frac{d^2 \sigma^2 \log T}{t}} \qquad \text{(since } \langle x^*, \theta^* \rangle \geq \langle x, \theta^* \rangle\text{)}$$

$$\geq \langle x, \widehat{\theta}_t \rangle - 8\sqrt{\frac{d^2 \sigma^2 \log T}{t}} \qquad \text{(using Lemma 27)}$$

Hence, the best arm will never satisfy the elimination criteria in Algorithm 1. $\qquad\square$

**Lemma 29** (Near optimality of Phase II arms)**.** *Suppose the hypothesis of Lemma 27 holds. Let $\tau = \frac{3072 d \sigma^2 \log T}{(\langle x^*, \theta^* \rangle)^2}$ Then, under the event $\mathcal{G}_2'$, at the beginning of each episode $\ell$, we have $\langle x, \theta^* \rangle \geq \langle x^*, \theta^* \rangle - 12\sqrt{\frac{3 d^2 \sigma^2 \log T}{2^\ell \cdot \tau}}$ for each arm $x$ in the surviving arm set $\widetilde{\mathcal{X}}$.*

*Proof.* Lemma 28 ensures that the optimal arm is contained in the surviving set of arms $\widetilde{\mathcal{X}}$. Furthermore, if an arm $x \in \widetilde{\mathcal{X}}$ is pulled in the $\ell^{\text{th}}$ episode, then it must be the case that arm $x$ was not eliminated in the previous episode (with a episode length parameter $\frac{T'}{2}$); in particular the arms $x$ does not satisfy the inequality on Line 4 of the second phase in Algorithm 1. This inequality reduces to

$$\langle x, \theta^* \rangle \geq \langle x^*, \theta^* \rangle - 8\sqrt{\frac{d^2 \sigma^2 \log T}{\frac{T'}{2}}} \geq \langle x^*, \theta^* \rangle - 12\sqrt{\frac{d^2 \sigma^2 \log T}{T'}}$$

Substituting $T' = 2^\ell \tau / 3$ in the above inequality proves the Lemma. $\qquad\square$

**Lemma 30** (Nash Welfare in LINPE)**.** *Let $\mathcal{G}' = \mathcal{G}_1 \cap \mathcal{G}_2'$. Suppose Assumption 1 holds. Then, under the event $\mathcal{G}$, the product of expected reward in LINPE satisfies*

$$\left(\prod_{t=\tau+1}^{T} \mathbb{E}[\langle x_t, \theta^* \rangle]\right)^{\frac{1}{T}} \geq \langle x^*, \theta^* \rangle^{\frac{T-\tau}{T}} \left(1 - 36\sqrt{\frac{d^2 \sigma^2}{T \langle x^*, \theta^* \rangle^2} \log T}\right)$$

*Proof.* Let $\mathcal{E}_j$ denote the time interval of the $j^{\text{th}}$ episode of LINPE, and $T_j'$ be the episode length. Recall that the LINPE

subroutine runs for at most $\log T$ episodes. Hence, from Lemma 29, we have under the event $\mathcal{G}'_2$,

$$\left(\prod_{t=\tau+1}^{T} \mathbb{E}[\langle x_t, \theta^* \rangle]\right)^{\frac{1}{T}} = \left(\prod_{\mathcal{E}_j} \prod_{t \in \mathcal{E}_j} \mathbb{E}[\langle x_t, \theta^* \rangle]\right)^{\frac{1}{T}} \geq \prod_{\mathcal{E}_j} \left(\langle x^*, \theta^* \rangle - 12\sqrt{\frac{d^2 \sigma^2 \log T}{T'_j}}\right)^{\frac{|\mathcal{E}_j|}{T}}$$

$$= \langle x^*, \theta^* \rangle^{\frac{T-\tau}{T}} \prod_{j=1}^{\log T} \left(1 - 12\sqrt{\frac{d^2 \sigma^2 \log T}{(\langle x^*, \theta^* \rangle)^2 T'_j}}\right)^{\frac{|\mathcal{E}_j|}{T}}$$

$$\geq \langle x^*, \theta^* \rangle^{\frac{T-\tau}{T}} \prod_{j=1}^{\log T} \left(1 - 12\sqrt{\frac{3 d^2 \sigma^2 \log T}{2(\langle x^*, \theta^* \rangle)^2 \tau}}\right)^{\frac{|\mathcal{E}_j|}{T}},$$

where the last inequality holds since $T'_j \geq \frac{2}{3}\tau$ for each episode $j$ of LINPE. Now, from Lemma 17, under the event $\mathcal{G}_1$, we have $\tau \geq \frac{864 d^2 \sigma^2 \log T}{(\langle x^*, \theta^* \rangle)^2}$ since by our design $p_a = 1$ when $p = 0$. This yields $12\sqrt{2\frac{d^2 \sigma^2 \log T}{2(\langle x^*, \theta^* \rangle)^2 \tau}} \leq 1/2$ and thus, invoking Fact 19, we get

$$\left(\prod_{t=\tau+1}^{T} \mathbb{E}[\langle x_t, \theta^* \rangle]\right)^{\frac{1}{T}} \geq \langle x^*, \theta^* \rangle^{\frac{T-\tau}{T}} \prod_{j=1}^{\log T} \left(1 - 24\frac{|\mathcal{E}_j|}{T}\sqrt{\frac{d^2 \sigma^2 \log T}{(\langle x^*, \theta^* \rangle)^2 T'_j}}\right)$$

$$\geq \langle x^*, \theta^* \rangle^{\frac{T-\tau}{T}} \prod_{j=1}^{\log T} \left(1 - 24\frac{T'_j + \frac{d(d+1)}{2}}{T}\sqrt{\frac{d^2 \sigma^2 \log T}{(\langle x^*, \theta^* \rangle)^2 T'_j}}\right) \qquad (|\mathcal{E}_j| \leq T'_j + \frac{d(d+1)}{2})$$

$$\geq \langle x^*, \theta^* \rangle^{\frac{T-\tau}{T}} \prod_{j=1}^{\log T} \left(1 - 36\frac{\sqrt{T'_j}}{T}\sqrt{\frac{d^2 \sigma^2 \log T}{(\langle x^*, \theta^* \rangle)^2}}\right) \qquad (\text{assuming } T'_j \geq d(d+1))$$

$$\geq \langle x^*, \theta^* \rangle^{\frac{T-\tau}{T}} \left(1 - \frac{36}{T}\sqrt{\frac{d^2 \sigma^2 \log T}{(\langle x^*, \theta^* \rangle)^2}} \sum_{j=1}^{\log T} \sqrt{T'_j}\right)$$

$$\geq \langle x^*, \theta^* \rangle^{\frac{T-\tau}{T}} \left(1 - \frac{36}{T}\sqrt{\frac{d^2 \sigma^2 \log T}{(\langle x^*, \theta^* \rangle)^2}} \sqrt{T \log T}\right) \qquad (\text{using Cauchy-Schwarz inequality})$$

$$\geq \langle x^*, \theta^* \rangle^{\frac{T-\tau}{T}} \left(1 - 36\sqrt{\frac{d^2 \sigma^2}{T(\langle x^*, \theta^* \rangle)^2}} \log T\right).$$

The proof concludes by noting that the above holds under the event $\mathcal{G}'$.[2] $\qquad\qquad\qquad\qquad \square$

## B.4. Nash Regret of FAIRLINBANDIT (Proof of Theorem 5)

We will give the proof for LINUCB. The proof for LINPE is identical (with only minor changes in constants). Without loss of generality, we assume that $\langle x^*, \theta^* \rangle \geq \sqrt{\frac{d^2 \sigma^2}{T}} \log T$. Otherwise, the Nash regret bound holds trivially. First note that under Assumption 1, the expected reward $\langle x, \theta^* \rangle \geq 0$ for all $x$. Hence $\mathbb{E}[\langle x_t, \theta^* \rangle] \geq \mathbb{E}[\langle x_t, \theta^* \rangle \mid \mathcal{G}] \cdot \Pr\{\mathcal{G}\}$ for all rounds $t$. Moreover, from Lemma 14 and 23, we have $\Pr\{\mathcal{G}_2\} \geq 1 - 4/T$.

---

[2]Note that the assumption $T'_j \geq d(d+1)$ can be guaranteed by initializing $T' = \max\{\frac{2}{3}\tau, d(d+1)\}$ in the subroutine LINPE. This implicitly assumes that $T > d(d+1)$, which is reasonable given that even the minimax optimal rates of $O(\frac{d}{\sqrt{T}})$ are vacuous for $T = \Omega(d^2)$. The same assumption is also used in Sawarni et al. (2023).

Now, from Lemma 20 and 26, we obtain

$$\left(\prod_{t=1}^T \mathbb{E}[\langle x_t, \theta^*\rangle]\right)^{\frac{1}{T}} \geq \left(\prod_{t=1}^\tau \mathbb{E}[\langle x_t, \theta^*\rangle]\right)^{\frac{1}{T}} \left(\prod_{t=\tau+1}^T \mathbb{E}[\langle x_t, \theta^*\rangle \mid \mathcal{G}] \cdot \Pr\{\mathcal{G}\}\right)^{\frac{1}{T}}$$

$$\geq \langle x^*, \theta^*\rangle \left(1 - \frac{\log(2(d+1))\tau}{T}\right) \left(1 - 16\sqrt{\frac{d^2\sigma^2}{T(\langle x^*, \theta^*\rangle)^2}} \log T\right) \Pr\{\mathcal{G}\}$$

$$\geq \langle x^*, \theta^*\rangle \left(1 - \frac{\log(2(d+1))\tau}{T} - 16\sqrt{\frac{d^2\sigma^2}{T(\langle x^*, \theta^*\rangle)^2}} \log T\right) \Pr\{\mathcal{G}\}$$

$$\geq \langle x^*, \theta^*\rangle \left(1 - \frac{\log(2(d+1))\tau}{T} - 16\sqrt{\frac{d^2\sigma^2}{T(\langle x^*, \theta^*\rangle)^2}} \log T\right) \left(1 - \frac{4}{T}\right)$$

$$\geq \langle x^*, \theta^*\rangle \left(1 - \frac{\log(2(d+1))d^2\sigma^2 \log T}{T(\langle x^*, \theta^*\rangle)^2} - 16\sqrt{\frac{d^2\sigma^2}{T(\langle x^*, \theta^*\rangle)^2}} \log(T) - \frac{4}{T}\right)$$

$$\geq \langle x^*, \theta^*\rangle - 16\sqrt{\frac{d^2\sigma^2}{T}} \log T - \frac{\log(2(d+1))d^2\sigma^2 \log T}{T\langle x^*, \theta^*\rangle} - \frac{4\langle x^*, \theta^*\rangle}{T}.$$

Hence, the Nash Regret can be bounded as

$$\mathrm{NR}_T = \langle x^*, \theta^*\rangle - \left(\prod_{t=1}^T \mathbb{E}[\langle x_t, \theta^*\rangle]\right)^{1/T} \leq 16\sqrt{\frac{d^2\sigma^2}{T}} \log T + \frac{\log(2(d+1))d^2\sigma^2 \log T}{T\langle x^*, \theta^*\rangle} + \frac{4\langle x^*, \theta^*\rangle}{T}$$

$$\leq 16\sqrt{\frac{d^2\sigma^2}{T}} \log T + \frac{\log(2(d+1))\sqrt{d^2\sigma^2}}{\sqrt{T}} + \frac{4}{T},$$

where the last inequality holds since $\langle x^*, \theta^*\rangle \geq \sqrt{\frac{d^2\sigma^2}{T}} \log T$ and $\langle x^*, \theta^*\rangle \leq 1$. Note that both these conditions can be simultaneously satisfied for a moderately large $T$ (depending on $\sigma, d$, e.g., $T \geq \widetilde{\Omega}(\sigma^2 d^2)$). Thus, the Nash Regret satisfies

$$\mathrm{NR}_T = O\left(\frac{\sigma d \log T}{\sqrt{T}}\right).$$

## C. $p-$mean Regret Analysis for FAIRLINBANDIT

We first state some standard results that will help us during the analysis.

**Lemma 31** (Generalized mean inequality). *Let $x_1, \dots, x_n \geq 0$ and for $r \in \mathbb{R}$ define*

$$M_r = \begin{cases} \left(\frac{1}{n} \sum_{i=1}^n x_i^r\right)^{1/r}, & r \neq 0, \\ \left(\prod_{i=1}^n x_i\right)^{1/n}, & r = 0. \end{cases}$$

*Then $M_r$ is strictly increasing in $r$; in particular, if $a < b$ then $M_a < M_b$.*

**Fact 32** (Sarkar et al. (2025)). *For all $q \geq 1$ and reals $x \in \left[0, \frac{1}{2q}\right]$, we have $(1-x)^{-q} \leq 1 + 2qx$.*

**Fact 33** (Sarkar et al. (2025)). *For all $0 < q \leq 1$ and reals $x \in \left[0, \frac{1}{2}\right]$, we have $(1-x)^{-q} \leq 1 + 2qx$.*

### C.1. Regret Bound for $p \geq 0$ (Proof of Theorem 7)

**Lemma 34** (p-mean regret of FAIRLINBANDIT for $p \geq 0$). *Consider a stochastic linear bandit problem with an infinite set of arms $\mathcal{X} \subset \mathbb{R}^d$ having $\sigma-$sub-gaussian rewards, time horizon $T$, and fairness parameter $p \in \mathbb{R}$. Then, the p-mean regret of FAIRLINBANDIT in the regime $p \geq 0$ satisfies*

$$\mathrm{R}_T^p \leq O\left(\frac{\sigma d \log T}{T}\right).$$

*Proof.* Invoking the generalized mean inequality (Lemma 31 for $a = 0$ and $b = p > 0$, we have

$$\left( \prod_{t=1}^{T} \mathbb{E}\left[ \langle x_t, \theta^* \rangle \right] \right)^{\frac{1}{T}} \leq \left( \frac{\sum_{t=1}^{T} (\mathbb{E}[\langle x_t, \theta^* \rangle])^p}{T} \right)^{\frac{1}{p}} .$$

Thus, we get the following bound on the p-mean regret

$$R_T^p \triangleq \langle x^*, \theta^* \rangle - \left( \frac{\sum_{t=1}^{T} (\mathbb{E}[\langle x_t, \theta^* \rangle])^p}{T} \right)^{\frac{1}{p}} \leq \langle x^*, \theta^* \rangle - \left( \prod_{t=1}^{T} \mathbb{E}\left[ \langle x_t, \theta^* \rangle \right] \right)^{\frac{1}{T}}$$

$$\leq 36 \sqrt{\frac{d^2 \sigma^2}{T}} \log(T) + \frac{\log(2(d+1)) \sqrt{d^2 \sigma^2}}{12\sqrt{T}} + \frac{6 \langle x^*, \theta^* \rangle}{T} \leq O\left( \frac{\sigma d \log T}{\sqrt{T}} \right) , \qquad \text{(using } \langle x^*, \theta^* \rangle \leq 1)$$

which completes the proof. □

### C.2. Regret Bound for $p < 0$ (Proof of Theorem 7)

In this case, we set $q = -p$ for notational convenience, so that we can use $|p_a| = |p| = q$ whenever required. Our objective then becomes analysing the following quantity (defined as the $q-$regret):

$$R_T^q \triangleq \langle x^*, \theta^* \rangle - \left( \frac{T}{\sum_{t=1}^{T} \frac{1}{(\mathbb{E}[\langle x_t, \theta^* \rangle])^q}} \right)^{\frac{1}{q}} .$$

Next, note that the algorithmic structure differs in Phase I and Phase II. Thus, to analyse these phases separately, we define

$$x \triangleq \frac{T}{\sum_{t=1}^{\tau} \frac{1}{\mathbb{E}[\langle x_t, \theta^* \rangle]^q}} \quad \text{and } y \triangleq \frac{T}{\sum_{t=\tau+1}^{T} \frac{1}{\mathbb{E}[\langle x_t, \theta^* \rangle]^q}}, \tag{13}$$

where $x$ and $y$ pertain to the rewards from Phase I and Phase II of FAIRLINBANDIT, respectively, so that we have

$$R_T^q = \langle x^*, \theta^* \rangle - \left( \frac{1}{\frac{1}{x} + \frac{1}{y}} \right)^{1/q} . \tag{14}$$

We will first focus on bounding $\frac{1}{x}$. The following lemma provides a bound for the same.

**Lemma 35.** *Under the event $\mathcal{G}$, $x$ satisfies* $\frac{1}{x} \leq \frac{\tau}{T} \left( \frac{2(d+1)}{\langle x^*, \theta^* \rangle} \right)^q$.

*Proof.* From Lemma 18, we have

$$\mathbb{E}[\langle x_t, \theta^* \rangle] \geq \mathbb{E}[\langle x_t, \theta^* \rangle \mid \mathcal{G}] \geq \frac{\langle x^*, \theta^* \rangle}{2(d+1)} \Leftrightarrow \frac{1}{(\mathbb{E}[\langle x_t, \theta^* \rangle])^q} \leq \left( \frac{2(d+1)}{\langle x^*, \theta^* \rangle} \right)^q .$$

Hence, we get

$$\frac{1}{x} \leq \frac{\tau}{T} \left( \frac{2(d+1)}{\langle x^*, \theta^* \rangle} \right)^q , \tag{15}$$

which completes the proof. □

Next, we focus on Phase II to bound $\frac{1}{y}$. Note that $\mathbb{E}[\langle x_t, \theta^* \rangle] \geq \Pr\{\mathcal{G}\} \mathbb{E}[\langle x_t, \theta^* \rangle \mid \mathcal{G}]$. Hence,

$$y \geq \frac{T}{\sum_{t=\tau+1}^{T} \frac{1}{\Pr\{\mathcal{G}\} \mathbb{E}[\langle x_t, \theta^* \rangle \mid \mathcal{G}]}} = \frac{T(\Pr\{\mathcal{G}\})^q}{\sum_{t=\tau+1}^{T} \frac{1}{(\mathbb{E}[\langle x_t, \theta^* \rangle \mid \mathcal{G}])^q}} . \tag{16}$$

Now, we know that by Jensen's inequality, $f(z) = z^{-\frac{1}{q}}$ is convex on $\mathbb{R}_{>0}$, for $q > 0$. Utilizing this result and the linearity of expectation, we get

$$\frac{1}{y} \leq \frac{\sum_{t=\tau+1}^{T} \frac{1}{(\mathbb{E}[\langle x_t, \theta^* \rangle | \mathcal{G}])^q}}{T(\Pr\{\mathcal{G}\})^q} \leq \frac{\sum_{t=\tau+1}^{T} \mathbb{E}\left[\frac{1}{\langle x_t, \theta^* \rangle^q} \middle| \mathcal{G}\right]}{T(\Pr\{\mathcal{G}\})^q} = \frac{\mathbb{E}\left[\sum_{t=\tau+1}^{T} \frac{1}{\langle x_t, \theta^* \rangle^q} \middle| \mathcal{G}\right]}{T(\Pr\{\mathcal{G}\})^q} . \tag{17}$$

Now, note that the two algorithms FAIRLINPE and FAIRLINUCB differ in Phase II. Hence, we analyze the quantity $y$ separately for the two algorithms. The following two lemmas establish the bounds on $\frac{1}{y}$.

**Lemma 36.** *Under the event $\mathcal{G}$, if the algorithm FAIRLINBANDIT calls LINUCB in Phase II, then $y$ satisfies*

$$\frac{1}{y} \leq \frac{1}{\left(\langle x^*, \theta^* \rangle^q - 16q \log T \langle x^*, \theta^* \rangle^{q-1} \sqrt{\frac{d^2 \sigma^2}{T}}\right) (\Pr\{\mathcal{G}\})^q} .$$

*Proof.* From Inequality (16), we have

$$\frac{1}{y} \leq \frac{\mathbb{E}\left[\sum_{t=\tau+1}^{T} \frac{1}{(\langle x_t, \theta^* \rangle)^q} \middle| \mathcal{G}\right]}{T(\Pr\{\mathcal{G}\})^q} \leq \frac{\mathbb{E}\left[\sum_{t=\tau+1}^{T} \left(\langle x^*, \theta^* \rangle - 2\beta_{t-1} \|x_t\|_{\overline{V}_{t-1}^{-1}}\right)^{-q} \middle| \mathcal{G}\right]}{T(\Pr\{\mathcal{G}\})^q} . \qquad \text{(via Lemma 24)}$$

Further, by the arguments in Lemma 25 and Lemma 26, we have $\|x_t\|_{\overline{V}_{t-1}^{-1}} \leq \sqrt{\frac{3d}{\tau}}$ and $\beta_{t-1} \leq \beta_T \leq \sqrt{4d\sigma^2 \log T}$. Hence, we get $2\beta_{t-1}\|x_t\|_{\overline{V}_{t-1}^{-1}} \leq \sqrt{\frac{48d^2\sigma^2 \log T}{\tau}}$. Now, we split our analysis into two cases: when $p < -1$ and $-1 \leq p < 0$ respectively.

*Case 1: $p < -1$:* In this case, $q = |p_a| = |p|$. Now, let $u = \sqrt{\frac{48d^2\sigma^2 \log T}{\tau}}$. Then we have

$$u = \sqrt{\frac{48d^2\sigma^2 \log T}{\tau}} \leq \sqrt{\frac{48d^2\sigma^2 \log T \langle x^*, \theta^* \rangle^2}{864p^2 d^2\sigma^2 \log T}} \leq \frac{\langle x^*, \theta^* \rangle}{2|p|} . \qquad (\tau \geq \tfrac{864p^2 d^2\sigma^2 \log T}{\langle x^*, \theta^* \rangle^2})$$

Thus, the quantity $x = \frac{1}{\langle x^*, \theta^* \rangle}\sqrt{\frac{48d^2\sigma^2 \log T}{\tau}} \leq \frac{1}{2|p|} = \frac{1}{2q}$ (as $|p| = q$).

*Case 2: $-1 \leq p < 0$:* For this case, $p_a = 1$. Again, let $u = \sqrt{\frac{48d^2\sigma^2 \log T}{\tau}}$. Then we have

$$u = \sqrt{\frac{48d^2\sigma^2 \log T}{\tau}} \leq \sqrt{\frac{48d^2\sigma^2 \log T \langle x^*, \theta^* \rangle^2}{864d^2\sigma^2 \log T}} \leq \frac{\langle x^*, \theta^* \rangle}{2} , \qquad (\tau \geq \tfrac{864p^2 d^2\sigma^2 \log T}{\langle x^*, \theta^* \rangle^2})$$

which implies, the quantity $x = \frac{1}{\langle x^*, \theta^* \rangle}\sqrt{\frac{48d^2\sigma^2 \log T}{\tau}} \leq \frac{1}{2}$. Thus, we have, by Facts 32 and 33 with $x = \frac{1}{\langle x^*, \theta^* \rangle}\sqrt{\frac{48d^2\sigma^2 \log T}{\tau}}, \forall p < 0$,

$$\frac{1}{y} \leq \frac{\mathbb{E}\left[\sum_{t=\tau+1}^{T} \langle x^*, \theta^* \rangle^{-q} \left(1 + \frac{4q\beta_{t-1}\|x_t\|_{\overline{V}_{t-1}^{-1}}}{\langle x^*, \theta^* \rangle}\right)\right]}{T(\Pr\{\mathcal{G}\})^q} \leq \langle x^*, \theta^* \rangle^{-q} \frac{T + \frac{4q\beta_T}{\langle x^*, \theta^* \rangle}\mathbb{E}\left[\sum_{t=\tau+1}^{T} \|x_t\|_{\overline{V}_{t-1}^{-1}}\right]}{T(\Pr\{\mathcal{G}\})^q} ,$$

where the last inequality holds using $\sum_{t=\tau+1}^{T} 1 \leq T$ and $\beta_{t-1} \leq \beta_T$. Now, using the analysis in Lemma 26, we know that $\mathbb{E}\left[\sum_{t=\tau+1}^{T} \|x_t\|_{\overline{V}_{t-1}^{-1}}\right] \leq \sqrt{4Td \log T}$ via Cauchy-Schwarz inequality. Thus, we have

$$\frac{1}{y} \leq \langle x^*, \theta^* \rangle^{-q} \frac{T + \frac{4q\beta_T \sqrt{4d \log T}}{\langle x^*, \theta^* \rangle}}{T(\Pr\{\mathcal{G}\})^q} \leq \frac{\langle x^*, \theta^* \rangle^{-q}\left(1 + \frac{16q \log T}{\langle x^*, \theta^* \rangle}\sqrt{\frac{d^2\sigma^2}{T}}\right)}{T(\Pr\{\mathcal{G}\})^q} \qquad \text{(Using } \beta_T \leq \sqrt{4d\sigma^2 \log T})$$

$$\leq \frac{1}{\left(\langle x^*, \theta^* \rangle^q - 16q \log T \langle x^*, \theta^* \rangle^{q-1}\sqrt{\frac{d^2\sigma^2}{T}}\right)(\Pr\{\mathcal{G}\})^q} ,$$

where the last inequality holds using $(1 + x) \leq \frac{1}{1-x} \ \forall \ 0 \leq x \leq 1$. This completes the proof of the lemma. $\qquad \square$

**Lemma 37.** *Under the event $\mathcal{G}$, if the algorithm* FAIRLINBANDIT *calls* LINPE *in Phase II, then $y$ satisfies*

$$\frac{1}{y} \leq \frac{1}{\left( \langle x^*, \theta^* \rangle^q - 48q \log T \langle x^*, \theta^* \rangle^{q-1} \sqrt{\frac{d^2 \sigma^2}{T}} \right) (\Pr\{\mathcal{G}\})^q} \; .$$

*Proof.* From Inequality (16), we have

$$\frac{\mathbb{E}\left[ \sum_{t=\tau+1}^{T} \frac{1}{(\langle x_t, \theta^* \rangle)^q} \Big| \mathcal{G} \right]}{T(\Pr\{\mathcal{G}\})^q} = \frac{\mathbb{E}\left[ \sum_{\varepsilon_j} \sum_{t \in \varepsilon_j} \frac{1}{\langle x_t, \theta^* \rangle^q} \Big| \mathcal{G} \right]}{T(\Pr\{\mathcal{G}\})^q} \leq \frac{\sum_{\varepsilon_j} |\varepsilon_j| \left( \langle x^*, \theta^* \rangle - 12 \sqrt{\frac{d^2 \sigma^2 \log T}{T'_j}} \right)^{-q}}{T(\Pr\{\mathcal{G}\})^q} \; . \quad \text{(via Lemma 29)}$$

Now, we split our analysis into two cases: when $p < -1$ and $-1 \leq p < 0$ respectively.

*Case 1: $p < -1$:* In this case, $q = |p_a| = |p|$. Now, let $u = 12\sqrt{\frac{d^2 \sigma^2 \log T}{T'_j}}$. Then we have

$$u = 12\sqrt{\frac{d^2 \sigma^2 \log T}{T'_j}} \leq 12\sqrt{\frac{3 d^2 \sigma^2 \log T}{2\tau}} \qquad (T'_j \geq \tfrac{2\tau}{3})$$

$$\leq 12\sqrt{\frac{d^2 \sigma^2 \log T \langle x^*, \theta^* \rangle^2}{576 p^2 d^2 \sigma^2 \log T}} = \frac{12 \langle x^*, \theta^* \rangle}{24|p|} \; . \qquad (\tau \geq \tfrac{864 p^2 d^2 \sigma^2 \log T}{\langle x^*, \theta^* \rangle^2})$$

Thus, the quantity $x = \frac{u}{\langle x^*, \theta^* \rangle} \leq \frac{1}{2|p|} = \frac{1}{2q}$ (as $|p| = q$).

*Case 2: $-1 \leq p < 0$:* For this case, $p_a = 1$. Again, let $u = 12\sqrt{\frac{d^2 \sigma^2 \log T}{T'_j}}$. Then we have

$$u = 12\sqrt{\frac{d^2 \sigma^2 \log T}{T'_j}} \leq 12\sqrt{\frac{3 d^2 \sigma^2 \log T}{2\tau}} \qquad (T'_j \geq \tfrac{2\tau}{3})$$

$$\leq 12\sqrt{\frac{d^2 \sigma^2 \log T \langle x^*, \theta^* \rangle^2}{576 d^2 \sigma^2 \log T}} = \frac{12 \langle x^*, \theta^* \rangle}{24} \; , \qquad (\tau \geq \tfrac{864 p^2 d^2 \sigma^2 \log T}{\langle x^*, \theta^* \rangle^2})$$

which implies, the quantity $x = \frac{u}{\langle x^*, \theta^* \rangle} \leq \frac{1}{2}$ . Thus, we have, by Facts 32 and 33 with $x = \frac{u}{\langle x^*, \theta^* \rangle}, \forall \, p < 0$,

$$\frac{1}{y} \leq \frac{\sum_{\varepsilon_j} |\varepsilon_j| \langle x^*, \theta^* \rangle^{-q} \left( 1 + \frac{24q}{\langle x^*, \theta^* \rangle} \sqrt{\frac{d^2 \sigma^2 \log T}{T'_j}} \right)}{T(\Pr\{\mathcal{G}\})^q}$$

$$\leq \langle x^*, \theta^* \rangle^{-q} \frac{\sum_{\varepsilon_j} |\varepsilon_j| + \sum_{\varepsilon_j} |\varepsilon_j| \left( \frac{24q}{\langle x^*, \theta^* \rangle} \sqrt{\frac{d^2 \sigma^2 \log T}{T'_j}} \right)}{T(\Pr\{\mathcal{G}\})^q}$$

$$\leq \langle x^*, \theta^* \rangle^{-q} \frac{T + \sum_{j=1}^{\log T} \left( T'_j + \frac{d(d+1)}{2} \right) \left( \frac{24q}{\langle x^*, \theta^* \rangle} \sqrt{\frac{d^2 \sigma^2 \log T}{T'_j}} \right)}{T(\Pr\{\mathcal{G}\})^q} \qquad (\sum_{\epsilon_j} |\varepsilon_j| \leq T)$$

$$\leq \langle x^*, \theta^* \rangle^{-q} \frac{T + \sum_{j=1}^{\log T} \left( 2T'_j \right) \left( 1 + \frac{24q}{\langle x^*, \theta^* \rangle} \sqrt{\frac{d^2 \sigma^2 \log T}{T'_j}} \right)}{T(\Pr\{\mathcal{G}\})^q} \qquad (\text{Assuming } T'_j \geq d(d+1) > d(d+1)/2)$$

$$\leq \frac{\langle x^*, \theta^* \rangle^{-q} \left( T + \sum_{j=1}^{\log T} \frac{48q \sqrt{T'_j}}{\langle x^*, \theta^* \rangle} \sqrt{d^2 \sigma^2 \log T} \right)}{T(\Pr\{\mathcal{G}\})^q} \qquad (\sum_j T'_j \leq T)$$

$$\leq \frac{\langle x^*, \theta^* \rangle^{-q} \left( T + \frac{48q \log T \sqrt{T}}{\langle x^*, \theta^* \rangle} \sqrt{d^2 \sigma^2} \right)}{T(\Pr\{\mathcal{G}\})^q} \qquad (\text{Cauchy-Schwarz Inequality})$$

$$= \frac{\langle x^*, \theta^* \rangle^{-q} \left( 1 + \frac{48q \log T}{\langle x^*, \theta^* \rangle} \sqrt{\frac{d^2 \sigma^2}{T}} \right)}{(\Pr\{\mathcal{G}\})^q} \leq \frac{1}{\left( \langle x^*, \theta^* \rangle^q - 48q \log T \langle x^*, \theta^* \rangle^{q-1} \sqrt{\frac{d^2 \sigma^2}{T}} \right) (\Pr\{\mathcal{G}\})^q} \; ,$$

where the last inequality holds using $(1+x) \leq \frac{1}{1-x} \ \forall \ 0 \leq x \leq 1$. This completes the proof of the lemma. $\qquad \square$

Now, we formally establish the regret bound for $p < 0$. We will give the proof for LINUCB. The proof for LINPE is identical (with only minor changes in constants).

**Lemma 38** (p-mean regret of FAIRLINBANDIT for $p < 0$). *Consider a stochastic linear bandit problem with an infinite set of arms $\mathcal{X} \subset \mathbb{R}^d$ having $\sigma-$sub-gaussian rewards, time horizon $T$, and fairness parameter $p \in \mathbb{R}$. Then, the p-mean regret of FAIRLINBANDIT in the regine $p < 0$ satisfies*

$$R_T^p \leq O\left(\frac{\sigma d^{\frac{|p|+2}{2}} \log T}{\sqrt{T}} \cdot \max(1, |p|)\right)$$

*Proof.* First, observe that when $\langle x^*, \theta^* \rangle \leq \frac{85|p|\sigma(2(d+1))^{\frac{|p|+2}{2}} \log T}{\sqrt{T}}$ the p-mean regret satisfies

$$R_T^q = \langle x^*, \theta^* \rangle - \left(\frac{\sum_{t=1}^{T}(\mathbb{E}[\langle x_t, \theta^* \rangle])^p}{T}\right)^{\frac{1}{p}} \leq \langle x^*, \theta^* \rangle \leq O\left(\frac{\sigma|p|d^{\frac{|p|+2}{2}}2^{\frac{|p|}{2}} \log T}{\sqrt{T}}\right). \tag{18}$$

Next, we consider the case when $\langle x^*, \theta^* \rangle \geq \frac{85|p|\sigma(2(d+1))^{\frac{|p|+2}{2}} \log T}{\sqrt{T}}$. Let $x$ and $y$ be the quantities as defined in Equation (13) and $q = -p$ as defined earlier. We focus on the quantity $\frac{1}{\frac{1}{x}+\frac{1}{y}}$. Then, using Lemma 35 and 36, we get

$$\frac{1}{\frac{1}{x}+\frac{1}{y}} \geq \frac{1}{\frac{\tau(2(d+1))^q}{\langle x^*, \theta^* \rangle^q T} + \frac{1}{\left(\langle x^*, \theta^* \rangle^q - 16q \log T \langle x^*, \theta^* \rangle^{q-1} \sqrt{\frac{d^2\sigma^2}{T}}\right)(\Pr\{\mathcal{G}\})^q}}$$

$$= \frac{\left(\langle x^*, \theta^* \rangle^q - 16q \log T \langle x^*, \theta^* \rangle^{q-1} \sqrt{\frac{d^2\sigma^2}{T}}\right)(\Pr\{\mathcal{G}\})^q}{1 + \frac{\tau(2(d+1))^q}{T}\left(1 - \frac{16q \log T}{\langle x^*, \theta^* \rangle}\sqrt{\frac{d^2\sigma^2}{T}}\right)(\Pr\{\mathcal{G}\})^q}.$$

Multiplying the numerator and denominator by $1 - \frac{\tau(2(d+1))^q}{T}\left(1 - \frac{16q \log T}{\langle x^*, \theta^* \rangle}\sqrt{\frac{d^2\sigma^2}{T}}\right)(\Pr\{\mathcal{G}\})^q$, we have

$$\frac{1}{\frac{1}{x}+\frac{1}{y}} \geq \frac{\left(\langle x^*, \theta^* \rangle^q - 16q \log T \langle x^*, \theta^* \rangle^{q-1} \sqrt{\frac{d^2\sigma^2}{T}}\right)(\Pr\{\mathcal{G}\})^q \left(1 - \frac{\tau(2(d+1))^q}{T}\left(1 - \frac{16q \log T}{\langle x^*, \theta^* \rangle}\sqrt{\frac{d^2\sigma^2}{T}}\right)(\Pr\{\mathcal{G}\})^q\right)}{1 - \left(\frac{\tau(2(d+1))^q}{T}\left(1 - \frac{16q \log T}{\langle x^*, \theta^* \rangle}\sqrt{\frac{d^2\sigma^2}{T}}\right)(\Pr\{\mathcal{G}\})^q\right)^2}$$

$$\geq \left(\langle x^*, \theta^* \rangle^q - 16q \log T \langle x^*, \theta^* \rangle^{q-1} \sqrt{\frac{d^2\sigma^2}{T}}\right)(\Pr\{\mathcal{G}\})^q \left(1 - \frac{\tau(2(d+1))^q}{T}\left(1 - \frac{16q \log T}{\langle x^*, \theta^* \rangle}\sqrt{\frac{d^2\sigma^2}{T}}\right)(\Pr\{\mathcal{G}\})^q\right),$$

where the last inequality holds because of the denominator being less than 1. We can further expand this to get

$$\frac{1}{\frac{1}{x} + \frac{1}{y}} \geq \langle x^*, \theta^* \rangle^q \left( 1 - \frac{16q \log T}{\langle x^*, \theta^* \rangle} \sqrt{\frac{d^2 \sigma^2}{T}} \right) (\Pr\{\mathcal{G}\})^q \left( 1 - \frac{\tau(2(d+1))^q}{T} \left( 1 - \frac{16q \log T}{\langle x^*, \theta^* \rangle} \sqrt{\frac{d^2 \sigma^2}{T}} \right) (\Pr\{\mathcal{G}\})^q \right)$$

$$\geq \langle x^*, \theta^* \rangle^q \log T \left( 1 - \frac{16q \log T}{\langle x^*, \theta^* \rangle} \sqrt{\frac{d^2 \sigma^2}{T}} \right) (\Pr\{\mathcal{G}\})^q \left( 1 - \frac{\tau(2(d+1))^q}{2T} \right)$$

$$\text{(since } \left( 1 - \frac{16q}{\langle x^*, \theta^* \rangle} \sqrt{\frac{d^2 \sigma^2 \log T}{T}} (\Pr\{\mathcal{G}\})^q \right) \leq 1)$$

$$\geq \langle x^*, \theta^* \rangle^q (\Pr\{\mathcal{G}\})^q \left( 1 - \frac{16q \log T}{\langle x^*, \theta^* \rangle} \sqrt{\frac{d^2 \sigma^2}{T}} - \frac{\tau(2(d+1))^q}{2T} \right)$$

$$\text{(using } (1-x)(1-y) \geq (1-x-y) \; \forall \, x, y > 0)$$

$$= \langle x^*, \theta^* \rangle^q (\Pr\{\mathcal{G}\})^q \left( 1 - \frac{16q \log T}{\langle x^*, \theta^* \rangle} \sqrt{\frac{d^2 \sigma^2}{T}} - \frac{1536 dS(2(d+1))^q}{T} \right) \qquad (\tau \leq 3072 dS)$$

$$= \langle x^*, \theta^* \rangle^q (\Pr\{\mathcal{G}\})^q \left( 1 - \frac{16q \log T}{\langle x^*, \theta^* \rangle} \sqrt{\frac{d^2 \sigma^2}{T}} - \frac{1536 q^2 d^2 \sigma^2 (2(d+1))^q \log T}{\langle x^*, \theta^* \rangle^2 T} \right) . \qquad (q = |p|)$$

Exponentiating the last inequality by $\frac{1}{q}$, we have

$$\left( \frac{1}{\frac{1}{x} + \frac{1}{y}} \right)^{\frac{1}{q}} \geq \langle x^*, \theta^* \rangle (\Pr\{\mathcal{G}\}) \left( 1 - \frac{16q \log T}{\langle x^*, \theta^* \rangle} \sqrt{\frac{d^2 \sigma^2}{T}} - \frac{1536 q^2 d^2 \sigma^2 (2(d+1))^q \log T}{\langle x^*, \theta^* \rangle^2 T} \right)^{\frac{1}{q}} .$$

Now, consider the following term

$$v = \frac{16q \log T}{\langle x^*, \theta^* \rangle} \sqrt{\frac{d^2 \sigma^2}{T}} + \frac{1536 q^2 d^2 \sigma^2 (2(d+1))^q \log T}{\langle x^*, \theta^* \rangle^2 T}$$

$$\leq \frac{16d}{85(2(d+1))^{\frac{q+2}{2}}} + \frac{1536}{7225} \leq \frac{1}{2} . \qquad (\langle x^*, \theta^* \rangle \geq \frac{85|p|\sigma(2(d+1))^{\frac{|p|+2}{2}} \log T}{\sqrt{T}})$$

We can apply Claim 19 on $(1-v)^{\frac{1}{q}}$ to get

$$\left( \frac{1}{\frac{1}{x} + \frac{1}{y}} \right)^{\frac{1}{q}} \geq \langle x^*, \theta^* \rangle (\Pr\{\mathcal{G}\}) \left( 1 - \frac{32 \log T}{\langle x^*, \theta^* \rangle} \sqrt{\frac{d^2 \sigma^2}{T}} - \frac{3072 q^2 d^2 \sigma^2 (2(d+1))^q \log T}{q \langle x^*, \theta^* \rangle^2 T} \right) .$$

Further, substituting $\Pr\{\mathcal{G}\} \geq 1 - \frac{4}{T}$ via union bound, we have

$$\left( \frac{1}{\frac{1}{x} + \frac{1}{y}} \right)^{\frac{1}{q}} \geq \langle x^*, \theta^* \rangle \left( 1 - \frac{4}{T} \right) \left( 1 - \frac{32 \log T}{\langle x^*, \theta^* \rangle} \sqrt{\frac{d^2 \sigma^2}{T}} - \frac{3072 q^2 d^2 \sigma^2 (2(d+1))^q \log T}{q \langle x^*, \theta^* \rangle^2 T} \right)$$

$$\geq \langle x^*, \theta^* \rangle \left( 1 - \frac{4}{T} - \frac{32 \log T}{\langle x^*, \theta^* \rangle} \sqrt{\frac{d^2 \sigma^2}{T}} - \frac{3072 q^2 d^2 \sigma^2 (2(d+1))^q \log T}{q \langle x^*, \theta^* \rangle^2 T} \right) .$$

$$\text{(using } (1-x)(1-y) \geq (1-x-y) \; \forall \, x, y > 0)$$

Thus, the $q$−regret satisfies

$$R_T^q \leq \langle x^*, \theta^* \rangle - \langle x^*, \theta^* \rangle \left( 1 - \frac{4}{T} - \frac{32 \log T}{\langle x^*, \theta^* \rangle} \sqrt{\frac{d^2 \sigma^2}{T}} - \frac{3072 q^2 d^2 \sigma^2 (2(d+1))^q \log T}{q \langle x^*, \theta^* \rangle^2 T} \right)$$

$$\leq \frac{4 \langle x^*, \theta^* \rangle}{T} + 32 \log T \sqrt{\frac{d^2 \sigma^2}{T}} + \frac{3072 q^2 d^2 \sigma^2 (2(d+1))^q \log T}{q \langle x^*, \theta^* \rangle T}$$

$$\leq \frac{4 \langle x^*, \theta^* \rangle}{T} + 32 \log T \sqrt{\frac{d^2 \sigma^2}{T}} + \frac{3072 \sigma (2(d+1))^{\frac{q+2}{2}}}{85 \sqrt{T}} . \qquad (\langle x^*, \theta^* \rangle \geq \frac{85|p|\sigma(2(d+1))^{\frac{|p|+2}{2}} \log T}{\sqrt{T}})$$

As a result, the $p-$mean regret satisfies

$$\mathrm{R}_T^p \leq O\left(\frac{\sigma d^{\frac{|p|+2}{2}}}{\sqrt{T}}\right) = O\left(\frac{\sigma d^{\frac{|p|+2}{2}}\log T}{\sqrt{T}}\right). \tag{19}$$

Thus, from inequalities (18) and (19), we get the final regret bound as

$$\mathrm{R}_T^p \leq O\left(\frac{\sigma d^{\frac{|p|+2}{2}}\log T}{\sqrt{T}} \cdot \max(1, |p|)\right),$$

which completes the proof of the lemma. □

## D. Computational Details

In this section, we present implementation details and some additional results from our numerical simulations.

**Computational complexity of Algorithm 1** We observe that Algorithm 1 (FAIRLINBANDIT) admits a polynomial-time implementation when number of arms is finite. Specifically, in Part I the algorithm invokes the subroutine PULLARMS (Algorithm 2) to compute the John Ellipsoid. Given the collection of arm vectors, this computation can be carried out efficiently (see Chapter 3 of (Todd, 2016)). Moreover, for the purposes of our algorithm, an approximate John Ellipsoid is sufficient, and such an approximation can be computed significantly faster (Cohen et al., 2019), namely in time $O(|\mathcal{X}|^2 d)$.

In addition, Part I requires solving the D-optimal design problem at most $O(\log T)$ times. This problem is a concave maximization task and can be efficiently addressed using the Frank-Wolfe algorithm with rank-1 updates. Each iteration of the method incurs a cost of $O(|\mathcal{X}|^2)$, and the total number of iterations is bounded by $O(d)$ (see, for example, Chapter 21 of (Lattimore & Szepesvári, 2020) and Chapter 3 of (Todd, 2016)). For Part II, the computational complexity of the LINPE algorithm (Algorithm 4) is $O(|\mathcal{X}|^2 d \log T + T)$, since the number of D-optimal design iterations is similarly bounded.

For LINUCB (Algorithm 3), each iteration requires $O(d^2)$ time for Line 3 and $O(|\mathcal{X}|)$ time for Line 4, leading to an overall complexity of $O(Td^2 + T|\mathcal{X}|)$. Putting everything together, we conclude that FAIRLINBANDIT runs in polynomial time under both implementations.

**Approximation of John Ellipsoid.** To approximate the center of John's ellipsoid efficiently, we compute the *Chebyshev center* of the convex hull of the active arm set. Let $\mathcal{P}_t$ denote the convex hull of the active arms at round $t$, represented as $\mathcal{P}_t = \{x \in \mathbb{R}^d \mid a_k^\top x \leq b_k, \ k = 1, \ldots, K\}$. The Chebyshev center is the center of the largest Euclidean ball inscribed within $\mathcal{P}_t$, found by solving the optimization problem:

$$\begin{aligned} \underset{c,\,r}{\text{maximize}} \quad & r \\ \text{s.t.} \quad & a_k^\top c + r\|a_k\|_2 \leq b_k, \quad \forall k \in [K]. \end{aligned}$$

This linear program acts as a computationally efficient surrogate for the nonlinear John ellipsoid problem and is solved using standard convex optimization solvers.

## E. Additional Experiments

**LinNash Baseline Comparison.** Experiments with Gaussian rewards in the paper show that our algorithms outperform the baseline LinNash (Sawarni et al., 2023). However, as per their theoretical framework, LinNash can't work under the Gaussian reward model. Thus, we further perform a set of experiments, strictly adhering to the sub-Poisson reward model required by LinNash, with a properly tuned the variance parameter. In particular, as described by Sawarni et al. (2023)), we scale and normalize arm embeddings so that the mean rewards lie in the range [0, 0.5], and then sample Bernoulli rewards using these embeddings. With this setup, we can choose $\nu = 1$, since a Bernoulli is 1-subpoisson (Lemma 1, Sawarni et al. 2023). Since Bernoulli is also sub-gaussian with $\sigma^2 = 0.25$, we set $\sigma^2 = 1$ in our algorithms for fairness in comparison. The following table shows the results with these new experiments, where we capture the obtained regret at different timesteps ($t$):

| Algorithm | Dataset | $t = 2*10^7$ | $t = 4*10^7$ | $t = 6*10^7$ | $t = 8*10^7$ | $t = 10^8$ |
|---|---|---|---|---|---|---|
| LinNash | Yahoo! LTRC | 0.368 | 0.293 | 0.249 | 0.223 | 0.207 |
| FairLinPE | Yahoo! LTRC | 0.337 | 0.235 | 0.179 | 0.148 | 0.126 |
| FairLinUCB | Yahoo! LTRC | 0.319 | 0.199 | 0.143 | 0.112 | 0.092 |
| LinNash | MSLR-WEB10K | 0.287 | 0.195 | 0.184 | 0.152 | 0.133 |
| FairLinPE | MSLR-WEB10K | 0.309 | 0.212 | 0.166 | 0.132 | 0.110 |
| FairLinUCB | MSLR-WEB10K | 0.301 | 0.184 | 0.132 | 0.103 | 0.084 |

**Ablation Studies.** We first compare our algorithms with variable and fixed length conditions. We choose Phase-I length ($\tilde{T}$) from the set $\{T/2, T/4, T/8\}$ as well as from Sawarni et al. (2023)'s recommendation: $\tilde{T} = 3d^{1.25}\sigma\sqrt{T \log T}$. We run FairLinPE for $T = 10^8$ rounds on the MSLR-WEB10K dataset under the Gaussian reward model. The following table shows the obtained regret under different settings:

| Algorithm | Dataset | Phase I Length | $t = 2*10^7$ | $t = 4*10^7$ | $t = 6*10^7$ | $t = 8*10^7$ | $t = 10^8$ |
|---|---|---|---|---|---|---|---|
| FairLinPE | MSLR-WEB10K | $T/2$ | 0.371 | 0.361 | 0.334 | 0.273 | 0.232 |
| FairLinPE | MSLR-WEB10K | $T/4$ | 0.370 | 0.273 | 0.200 | 0.161 | 0.134 |
| FairLinPE | MSLR-WEB10K | $T/8$ | 0.276 | 0.178 | 0.141 | 0.121 | 0.108 |
| FairLinPE | MSLR-WEB10K | $3d^{1.25}\sigma\sqrt{T \log T}$ | 0.126 | 0.118 | 0.116 | 0.115 | 0.112 |
| FairLinPE | MSLR-WEB10K | Variable | 0.133 | 0.111 | 0.085 | 0.077 | 0.069 |

As shown in the table, the variable-stopping condition (proposed method) yields much better Nash regret performance than fixed-stopping rules.

Next, we perform experiments with different Phase I warmup strategies: a mixture of John Ellipsoid and D-Optimal Design (proposed method), only John Ellipsoid, and only D-Optimal Design. The following table presents the results for these experiments:

| Algorithm | Dataset | Phase I Strategy | $t = 2*10^7$ | $t = 4*10^7$ | $t = 6*10^7$ | $t = 8*10^7$ | $t = 10^8$ |
|---|---|---|---|---|---|---|---|
| FairLinPE | MSLR-WEB10K | D-Optimal | 0.175 | 0.113 | 0.086 | 0.076 | 0.069 |
| FairLinPE | MSLR-WEB10K | John Ellipsoid | 0.182 | 0.116 | 0.089 | 0.080 | 0.071 |
| FairLinPE | MSLR-WEB10K | Mixture | 0.173 | 0.111 | 0.085 | 0.073 | 0.066 |
| FairLinPE | Yahoo! LTRC | D-Optimal | 0.086 | 0.074 | 0.065 | 0.057 | 0.052 |
| FairLinPE | Yahoo! LTRC | John Ellipsoid | 0.088 | 0.072 | 0.064 | 0.055 | 0.051 |
| FairLinPE | Yahoo! LTRC | Mixture | 0.068 | 0.062 | 0.057 | 0.048 | 0.041 |

As shown in the table, using a mixture of John Ellipsoid and D-Optimal Design yields better regret performance compared to a simpler warm-up strategy that uses only one of them.

Finally, to empirically validate the effect of increasing the number of arms, we evaluate our algorithm in a synthetic linear bandit environment, systematically varying the number of arms $N \in \{10^2, 10^3, 10^4\}$. For each configuration, we fix the feature dimension at $d = 10$ and generate a ground-truth parameter $\theta^*$ by training a Lasso model on $n = 2000$ noisy synthetic observations. This training step ensures that $\theta^*$ possesses a sparse structure, mimicking the feature selection process used in our MSLR and Yahoo! experiments. The arm features are sampled from a standard normal distribution and subsequently normalized to the unit sphere. We then flip any arms with a mean reward $\langle x, \theta^* \rangle < 0$, ensuring strictly positive rewards across the action space. As illustrated in Figures 3(a) and 3(b), the algorithm exhibits robust performance across varying arm densities, with regret increasing with the number of arms.

# F. On the reward model assumed in Sawarni et al. (2023)

Sawarni et al. (2023) considered a very specific reward model to satisfy the technical necessity of using multiplicative concentration bounds in line with their use of estimate-dependent confidence widths. Their reward model has only non-negative rewards and follows a $\nu$-sub-Poisson distribution, which yields a specific multiplicative concentration bound. This reward model is restrictive, as it admits only non-negative rewards, unlike the more general sub-Gaussian rewards. Sawarni

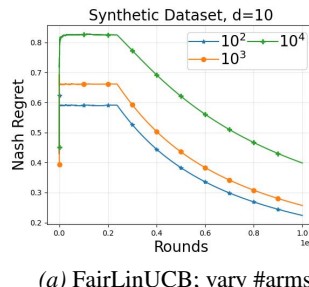

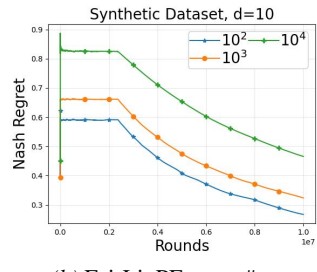

*(a)* FairLinUCB; vary #arms  *(b)* FairLinPE; vary #arms

*Figure 3.* Numerical results showing the effect of variation of the number of arms on the Nash regret. (a) and (b) show the effect on FairLinPE and FairLinUCB, respectively.

et al. (2023, lemma 2) shows that if the random reward $r_x$ is non-negative and $\sigma$-sub-Gaussian with mean $\langle x, \theta^* \rangle$, then it is also $\nu = \frac{\sigma^2}{\langle x, \theta^* \rangle}$ sub-Poisson. Hence, one would expect their analysis to hold for non-negative sub-Gaussian rewards as well. However, upon closer inspection, one would realize this is not the case. Recall that the analysis of their LINNASH algorithm (which uses a fixed number of rounds $\widetilde{T}$ for Part I) hinges crucially on estimate-dependent confidence widths

$$\left| \langle x, \theta^* \rangle - \langle x, \widehat{\theta} \rangle \right| \leq 3 \sqrt{\frac{d^2 \, \nu \langle x, \theta^* \rangle \, \log T}{\widetilde{T}}} \ .$$

Now, if we put $\nu = \frac{\sigma^2}{\langle x, \theta^* \rangle}$, then the confidence widths would become estimate-independent, leading to failure of their analysis. Thus, while the sub-Poisson rewards admit non-negative sub-Gaussian rewards as a special case, the value of the corresponding sub-Poisson parameter renders this observation entirely unhelpful for analyzing their algorithm.

