# OpenReview forum: "Improved Algorithms for Nash Welfare in Linear Bandits"
_ICML.cc/2026/Conference — ICML 2026 regular_

### Official Review · Reviewer_3a4d · 2026-03-08

**Soundness:** 3
**Presentation:** 3
**Significance:** 3
**Originality:** 3
**Overall Recommendation:** 4
**Confidence:** 3

**Summary:**

This paper studies the Nash welfare, which uses geometric mean of rewards to ensure a fairer metric compared to traditional arithmetic mean. Compared to the state of the art that has Nash regret $\tilde{O}(d^{5/4}/\sqrt{T})$, this paper proposes a FairLinBandit framework using Phased Elimination and UCB. A near-optimal regret is shown $O(d\log(T)/\sqrt{T})$, which matches the known lower bound. The paper also initiates the study of p-mean regrets. Experiments on real-world data complement the theory.

**Compliance With Llm Reviewing Policy:**

Affirmed.

**Final Justification:**

I do not have other concerns.

**Key Questions For Authors:**

1. Please see the weakness part.

2. In Figure 1(c), the proposed algorithms behave worse than the baseline (LinNash). Do you have an explanation?

3. Since the Nash objective of the paper is still to identify an optimal arm, I am wondering whether standard MAB algorithms(e.g. UCB) can achieve a sub-linear regret. If not, can you explain why these algorithms fail intuitively?

**Limitations:**

yes

**Strengths And Weaknesses:**

Strengths:
1. The paper proposes a novel FairLinBandit algorithm that imporves the regret bound compared to the state of the art. The regret bound is also near-optimal to the lower bound.
2. The introduction of p-means regret is a generalized framework that allows a tunable trade-off between fairness (Nash regret) and utility (arithmetic mean).
3. The experiments are extensive with details including the performance comparison to previous algorithms and computational cost comparison.

Overall, I believe this is a solid paper with strong contributions.

Weaknesses:
1. However, I am confused of the formulation of the Nash regret in single-agent setting. Compared to the Nash social welfare in the multi-agent setting (e.g. https://arxiv.org/abs/2007.06699) whose objective is a fair allocation among agents, this paper seems to achieve a 'fair' distribution across rounds. The objective is still an optimal arm. Therefore, the relation to the fairness is not clear.

---

> ### Author Rebuttal · Authors · 2026-03-28
>
> We thank the reviewer for their positive review and the thoughtful conceptual questions.
>
> * **Nash Regret in Single-Agent Settings:** In stochastic bandits, "fairness across rounds" translates to fairness among individuals in settings where each round represents a distinct user (e.g., loan approvals, sequential medical treatments). Here, fairness ensures that no user (e.g., round) is severely penalized. The objective is indeed to identify the optimal arm, but the *path* taken matters. Nash regret penalizes policies that pull arms with very low expected rewards during the learning phase, ensuring that the "worst-off" users in the sequence are protected from severely suboptimal decisions.
> * **Why Standard UCB Fails:** Standard MAB algorithms fail to achieve sub-linear Nash regret because they are purely utilitarian. Consider expected rewards bounded in [0, 1]. UCB might pull an arm, say, with an expected reward of 0.01, many times simply to shrink its confidence interval. In standard average regret, a few pulls of a 0.01 arm are easily absorbed by later pulls of the optimal arm (say, with an expected reward of 0.99). However, in Nash regret, a single near-zero expected reward severely punishes the geometric mean. Our algorithm succeeds by using Phase I to ensure no near-zero arms are ever pulled by UCB in Phase II.
> * **Figure 1(c) Behavior:** In Figure 1(c) (which tests the shorter horizon $T=10^7$), LinNash initially appears to have lower Nash regret. This happens because LinNash relies on overly conservative, estimate-dependent confidence intervals. These wide intervals force it to explore heavily and uniformly early on, which inadvertently keeps the geometric mean artificially high. However, this over-exploration prevents it from exploiting effectively. As time progresses, FairLinPE and FairLinUCB successfully identify the optimal arm, and their regret drops sharply, whereas LinNash remains unstable because it cannot sufficiently narrow its confidence bounds for shorter horizons ($T=10^7$).

---

> > ### Author Rebuttal · Reviewer_3a4d · 2026-04-02
> >
> > My concern about single-agent setting is addressed. I encourage the authors to include the discussion in their paper.

---

### Official Review · Reviewer_w7Tq · 2026-03-09

**Soundness:** 3
**Presentation:** 2
**Significance:** 3
**Originality:** 3
**Overall Recommendation:** 4
**Confidence:** 3

**Summary:**

This paper studies the problem of minimizing Nash regret in stochastic linear bandits. The proposed `FairLinBandit` algorithm achieves an order-optimal Nash regret bound and outperforms existing methods. The authors also introduce the notion of $p$-means regret in linear bandits and extend the analysis to more general sub-Gaussian reward distributions, which allows the framework to cover negative values of $p$.

**Compliance With Llm Reviewing Policy:**

Affirmed.

**Key Questions For Authors:**

In Figure 1(c), both `FairLinPE` and `FairLinUCB` still perform worse than the baseline algorithm `LinNash`, even a little bit after more than half of the rounds. What causes this? Does this suggest that `FairLinPE` and `FairLinUCB` have a slower convergence rate in practice?

**Limitations:**

See the section Weaknesses and Questions.

**Strengths And Weaknesses:**

**Strengths**: The paper is well motivated and addresses an open problem in the literature. The proposed algorithm improves Nash welfare in linear bandits.  I like the idea of p-means regret, which generalizes Nash regret.

**Weaknesses**:
1. Although the technical development is solid, the overall contribution appears incremental relative to existing work.
2. The organization of the paper could be improved. For example, placing the related works section in the main body may make the structure easier to follow.

---

> ### Author Rebuttal · Authors · 2026-03-28
>
> Thank you for the constructive feedback. We will follow your suggestion and move the related works section into the main body to improve the narrative flow (since one extra page is typically allowed in the final version).
>
> * **Figure 1(c) Convergence Behavior:** In Figure 1(c) (which tests the shorter horizon $T=10^7$), LinNash initially appears to have lower Nash regret. This happens because LinNash relies on overly conservative, estimate-dependent confidence intervals. These wide intervals force it to explore heavily and uniformly early on, which inadvertently keeps the geometric mean artificially high. However, this over-exploration prevents it from exploiting effectively. As time progresses, FairLinPE and FairLinUCB successfully identify the optimal arm, and their regret drops sharply, whereas LinNash remains unstable because it cannot sufficiently narrow its confidence bounds for shorter horizons ($T=10^7$).

---

> > ### Author Rebuttal · Reviewer_w7Tq · 2026-04-04
> >
> > Ack

---

### Official Review · Reviewer_NEX4 · 2026-03-12

**Soundness:** 3
**Presentation:** 4
**Significance:** 3
**Originality:** 3
**Overall Recommendation:** 5
**Confidence:** 3

**Summary:**

This work deals with the problem of achieving optimal Nash regret in Linear bandits by designing efficient algorithmic strategy that also utility-fairness tradeoff with phase transitions. It positively addresses an open problem in the existing literature (baseline). The authors argue that failure of existing approaches does not stem from the complexity of the problem, rather lack of efficient algorithm design. This work, with sufficient background on the problem, starts with sub-Gaussian reward structure to propose a meta algorithm FairLinBandit that has two phases. In phase 1, or the *exploration phase* it plays arms uniformly either from a G-Optimal design (helps in warming up estimate of the shared reward parameter $\theta$) and an ellipsoid based strategy (bounds estimated arm rewards away from zero). Termination of this exploration phase is decided by a novel problem dependent stopping rule devised in this work. In Phase 2, or the regret minimization phase,  FairLinBandit leverage standard bandit algorithms like UCB and PhasedElimination (PE). The authors provide a comprehensive theoretical analysis, later validated by sufficient empirical evidence.

**Compliance With Llm Reviewing Policy:**

Affirmed.

**Final Justification:**

I was impressed by this work from the beginning. I think the work is timely, bridges an important gap between bandits and fairness. The contributions are novel, unique. After the rebuttal by the authors, my concern were resolved, reinforcing my initial evaluation. Thus, I would like to advocate for a positive evaluation for this submission.

**Key Questions For Authors:**

- From the theoretical guaranties, UCB and PE as regret minimizing algorithms incur similar regret upper bound. Though empirically I observe (also mentioned by authors) that UCB outperforms PE almost consistently. What is the rationale behind using and keeping both the algorithms in the paper?

- Can you give an example where in this context arms can act as agents? Are they users, tasks, objectives?

- Do matching lower bound exists in this context? Can you discuss on the tightness of the regret upper bound?

- Most linear bandits use optimism in the face of uncertainty. As per my understanding, the nature of optimism should change for non-linear Nash welfare. How does it change exactly? Also, does FairLinBandit solve optimisation for Nash welfare per step? Is it essentially a non-convex optimisation? I may have not understood, but do you consider some approximations?

- Just for curiosity, how easy it is to adapt FairLinBandit for Multi-objective (linear) bandits? Is it just a specific scalarization?

**Limitations:**

Yes.

**Strengths And Weaknesses:**

- **Strengths:** The paper positively solve extension (subGaussian rewards, p-mean regret etc.) of an open problem in the existing literature on Nash regret in Linear bandits, thus very well-motivated. Being a theory heavy paper, it is very well presented. It provides sufficient background and intuition on background of the problem that helped me as a reader connect this paper easily to the existing works. The phase transitions in utility-fairness tradeoffs is clean and intuitive. I think significance of this problem lies in the fact that often regret minimizing bandit algorithm that are designed to solely on average regret does not reflect fairness notions, which becomes imperative in responsible AI in general.


- **Weakness:** See Questions below.

---

> ### Author Rebuttal · Authors · 2026-03-28
>
> We thank the reviewer for their strong support of the paper and for recognizing the value of the phase transitions.
>
> * **UCB vs. PE:** We included both to demonstrate that FairLinBandit acts as a generic meta-algorithm that can plug into any standard linear bandit solver. While UCB empirically outperforms PE (because it updates estimates at every step rather than at the end of each epoch), PE has lower time complexity (because it restricts arm selection to the shrinking set of good arms). See Remark 4 in the paper.
> * **Arms as Agents Context:** A classic example is sequential clinical trials or resource allocation. The "arm" is the treatment protocol or resource, and the "agent" is the distinct patient or demography receiving it at round $t$. Nash regret ensures that the algorithm does not heavily starve a specific patient or demography of effective treatments just to exploit a slightly better overall average.
> * **Lower Bounds and Tightness:** The standard average regret lower bound for linear bandits is $\Omega(d/\sqrt{T})$. By the AM-GM inequality, Nash regret is strictly lower-bounded by average regret. Therefore, our upper bound of $\widetilde{\mathcal{O}}(d/\sqrt{T})$ matches the information-theoretic lower bound up to logarithmic factors (see Remark 6 in the paper). For $p$-means regret, our upper bounds are optimal in $T$, but whether they are optimal in $d$ is an open question (see the discussion after Theorem 7 in the paper).
> * **Nature of Optimism:** FairLinBandit does not solve a non-convex optimization per step. Instead, it alters the *initialization* for standard optimistic algorithms. Standard UCB fails for Nash welfare because initial optimism might pull an arm with a true reward near zero, severely punishing the geometric mean. Our Phase I prevents this by robustly warming up the estimates, so that standard UCB (in Phase II) never overestimates so wildly as to pull a near-zero reward arm.
> * **Multi-objective Adaptation:** Extending to multi-objective bandits is an interesting direction. It would likely involve a scalarization approach, where the $p$-means objective is applied across the multiple reward vectors, though the concentration bounds might require either a union bound over the objectives or a bound on the maximum.

---

> > ### Author Rebuttal · Reviewer_NEX4 · 2026-04-03
> >
> > I thank the authors for their response. My questions are adequately addressed. I remain positive about this paper.

---

### Official Review · Reviewer_vzDm · 2026-03-13

**Soundness:** 3
**Presentation:** 2
**Significance:** 3
**Originality:** 3
**Overall Recommendation:** 4
**Confidence:** 3

**Summary:**

This submission studies fairness-aware linear bandits via Nash regret and a broader p-means regret objective, and proposes a two-phase meta-algorithm, FairLinBandit, with LINUCB/LINPE instantiations. The paper claims resolving an open problem from NeurIPS 2023 (Sawarni et al.) by eliminating the suboptimal $O(d^{5/4})$ dimension dependence and achieving the order-optimal $\tilde{O}(d/\sqrt{T})$ Nash regret bound. The high level idea is a data-adaptive stopping rule during an initial pure-exploration phase, which circumvents the need for restrictive, estimate-dependent "Nash Confidence Bounds" (NCB) and enables the use of tighter, standard UCB intervals.

**Compliance With Llm Reviewing Policy:**

Affirmed.

**Final Justification:**

The rebuttal addressed my concerns. I'm raising my score.

**Key Questions For Authors:**

Will you provide new experiments to fairly compare with LINNASH (strictly non-negative rewards, properly tuned $\nu$)?

**Limitations:**

Yes

**Strengths And Weaknesses:**

Strengths
- The topic is important. Closing the gap from $O(d^{5/4})$ to $O(d)$ in ambient dimension dependence is an important theoretical result.
- The high-level idea — a warm-up exploration phase followed by a standard optimistic linear bandit method — is very sound.
- Extending from Nash regret to p-means regret is conceptually interesting.

Weaknesses.
- Theoretical:
    - Missing assumptions: The authors explicitly define $\mathcal{X}$ as a countably infinite set. However, a countably infinite set bounded in an $L_2$ norm is generally not compact (it may not contain its limit points), meaning the optimal arm $x^*$ might not exist (the supremum may be unattainable). Furthermore, if $\mathcal{X}$ does not span $\mathbb{R}^d$, meaning a full-dimensional John Ellipsoid might not exist, and the $\frac{1}{d+1}$ volume guarantee goes away. The paper invokes Kiefer–Wolfowitz only under the stronger condition that X is compact and spans $\mathbb{R}^d$.
    - To prove the matrix inequality $V_t \succeq \frac{t}{3} U(\lambda)$, the authors explicitly claim (Line 684) that every arm $z$ in the support has been pulled at least $\lceil \lambda_z t/3 \rceil$ times. Because $\sum \lambda_z = 1$, the sum of the exact fractional values is exactly $t/3$. However, because of the ceiling function, the sum is strictly pushed higher. In the worst case, the required pulls equal $t/3 + \text{size}(\text{Supp}) - 1$, which can be up to $t/3 + d(d+1)/2$. Lemma 11 only guarantees that the arms from D-optimal design have been pulled at least $t/3$ w.h.p., and that doesn’t cover the additive term $d(d+1)/2$.
    - Algorithm 1 uses a doubling schedule and then sets $\tau = \tilde{T} / 2$ at the handoff to Phase II, i.e. essentially the last epoch length. But the text/proof implies $\tau$ is the total number of Phase-I rounds. I think $\tau$ should be $\tilde{T}$ in the pseudocode.
    - Algorithm 2 initializes counts for all $z \in X$ which is impossible since $X$ is is countably infinite.
    - Lemma 34 states $R_T^p \le O(\sigma d\log T/T)$. I think this should be $R_T^p \le O(\sigma d\log T/ \sqrt{T})$
    - Assumption 1 only requires nonnegative expected rewards, but for $p<0$ the proof uses inverse powers which requires strict positivity.

I think these issues are fixable, but overall the math has to be more precise.

- Experimental
    - Comparison to LINNASH is not fair: The experiments use Gaussian rewards $r_t=\langle x_t,\theta^\*\rangle+\eta_t$ with $\eta_t\sim\mathcal N(0,\sigma^2)$ , while the baseline LINNASH is explicitly described as assuming a different reward model (nonnegative / sub-Poisson-type). The paper even says “we choose $\nu=1$ for LINNASH. That is not a convincing or principled calibration. Because LINNASH assumes a sub-Poisson reward distribution, it requires a tuning parameter $\nu$ to scale the width of its Upper Confidence Bounds (UCBs).
    - The “real-world” experiments are not truly real-world bandit evaluations. The datasets are used to create a highly synthetic non-contextual linear environment.
    - The paper’s main novelty is the data-adaptive stopping rule and the reduction to standard linear bandit algorithms. Yet there is no direct ablation showing:
        * adaptive stopping vs fixed exploration length
        * John/D-optimal mixture vs simpler warm-start strategies

---

> ### Author Rebuttal · Authors · 2026-03-28
>
> We thank the reviewer for their meticulous review.
>
> * **Mathematical Precision & Assumptions:** We will implement the suggested technical corrections. First, we will update Assumption 1 to explicitly require $\mathcal{X}$ to be a compact subset spanning $\mathbb{R}^d$ (resolving the Kiefer-Wolfowitz invocation, a requirement also shared by Sawarni et al., 2023) and state the strict positivity needed for $p < 0$. Second, we will update Lemmas 11 and 17 to cleanly absorb the worst-case $d(d+1)/2$ ceiling function term into the $\widetilde{\Theta}(d^2)$ Phase I length. Note that since this is an additive constant independent of $T$, our asymptotic $\widetilde{\mathcal{O}}(d/\sqrt{T})$ regret remains completely intact. Finally, we will correct the minor typos noted in Algorithms 1 and 2 and Lemma 34.
>
> * **LinNash Baseline Comparison:** Experiments with Gaussian rewards in the paper show that our algorithms outperform the baseline LinNash (Sawarni et al., 2023). However, you are correct in pointing out that, theoretically, LinNash can't work under the Gaussian reward model. So we have run a new set of experiments, strictly adhering to the sub-Poisson reward model required by LinNash, and properly tuning the variance parameter $\nu$. In particular, as described by Sawarni et al. (2023), we scale and normalize arm embeddings so that the mean rewards lie in the range [0, 0.5], and then sample Bernoulli rewards using these embeddings. With this setup, we can choose $\nu = 1$, since a Bernoulli is 1-subpoisson (Lemma 1, Sawarni et al. 2023). Since Bernoulli is also sub-gaussian with $\sigma^2=0.25$, we set $\sigma^2=1$ in our algorithms for fairness in comparison. The following table shows the results with these new experiments, where we capture the obtained regret at different timesteps ($t$):
>
> | Algorithm | Dataset | $t=2*10^7$ | $t=4*10^7$ | $t=6*10^7$ | $t=8*10^7$ | $t=10^8$ |
> |---|---|---|---|---|---|---|
> | LinNash | Yahoo! LTRC | 0.368 | 0.293 | 0.249 | 0.223 | 0.207 |
> | FairLinPE | Yahoo! LTRC | 0.337 | 0.235 | 0.179 | 0.148 | 0.126 |
> | FairLinUCB | Yahoo! LTRC | 0.319 | 0.199 | 0.143 | 0.112 | 0.092 |
> | LinNash | MSLR-WEB10K | 0.287 | 0.195 | 0.184 | 0.152 | 0.133 |
> | FairLinPE | MSLR-WEB10K | 0.309 | 0.212 | 0.166 | 0.132 | 0.110 |
> | FairLinUCB | MSLR-WEB10K | 0.301 | 0.184 | 0.132 | 0.103 | 0.084 |
>
> The new results continue to demonstrate that FairLinUCB and FairLinPE converge faster and remain more stable than LinNash, even under its own assumed reward structure. We will make two plots from this table and put them in the paper (leveraging the extra page in the final version).
>
> * **Ablation Studies:** As suggested, we first compare our algorithms with variable and fixed length conditions. We choose Phase-I length ($\widetilde{T}$) from the set {$T/2$, $T/4$, $T/8$} as well as from Sawarni et al. recommendation: $\widetilde{T} = 3d^{1.25}\sigma\sqrt{T\log T}$. We run FairLinPE for $T=10^8$ rounds on the MSLR-WEB10K dataset under the Gaussian reward model. The following table shows the obtained regret under different settings:
>
> | Phase I Length | $t=2*10^7$ | $t=4*10^7$ | $t=6*10^7$ | $t=8*10^7$ | $t=10^8$ |
> |---|---|---|---|---|---|
> | $T/2$ | 0.371 | 0.361 | 0.334 | 0.273 | 0.232 |
> | $T/4$ | 0.370 | 0.273 | 0.200 | 0.161 | 0.134 |
> | $T/8$ | 0.276 | 0.178 | 0.141 | 0.121 | 0.108 |
> | $3d^{1.25}\sigma\sqrt{T \log T}$ | 0.126 | 0.118 | 0.116 | 0.115 | 0.112 |
> | Variable ( Ours) | 0.133 | 0.111 | 0.085 | 0.077 | 0.069 |
>
> As shown in the table, the variable-stopping condition (our proposed method) yields much better Nash regret performance than fixed-stopping rules.
>
> Next, we perform experiments with different Phase I warmup strategies: a mixture of John Ellipsoid and D-Optimal Design (proposed method), only John Ellipsoid, and only D-Optimal Design. The following table presents the results of FairLinPE for these experiments:
>
> | Dataset | Phase I Strategy | $t=2*10^7$ | $t=4*10^7$ | $t=6*10^7$ | $t=8*10^7$ | $t=10^8$ |
> |---|---|---|---|---|---|---|
>  | MSLR-WEB10K | D-Optimal | 0.175 | 0.113 | 0.086 | 0.076 | 0.069 |
>  | MSLR-WEB10K | John Ellipsoid | 0.182 | 0.116 | 0.089 | 0.080 | 0.071 |
>  | MSLR-WEB10K | Mixture | 0.173 | 0.111 | 0.085 | 0.073 | 0.066 |
>  | Yahoo! LTRC | D-Optimal | 0.086 | 0.074 | 0.065 | 0.057 | 0.052 |
>  | Yahoo! LTRC | John Ellipsoid | 0.088 | 0.072 | 0.064 | 0.055 | 0.051 |
>  | Yahoo! LTRC | Mixture | 0.068 | 0.062 | 0.057 | 0.048 | 0.041 |
>
> As shown in the table, using a mixture of John Ellipsoid and D-Optimal Design yields better regret performance compared to a simpler warm-up strategy that uses only one of them. Similar results are observed for FairLinUCB. We will put these results in the appendix of the paper.

---

> > ### Author Rebuttal · Reviewer_vzDm · 2026-04-04
> >
> > I thank the authors for their response. My concerns have been addressed.

---

### Decision · Program_Chairs · 2026-04-30

**Decision:**

Accept (regular)

**Comment:**

This work focused fairness-aware linear bandits via Nash regret and a broader p-means regret objective, and proposes a two-phase meta-algorithm, FairLinBandit, with LINUCB/LINPE instantiations. The paper claimed resolving an open problem from NeurIPS 2023 (Sawarni et al.) and provided a new algorithm with upper bound optimal up to log factors. The contribution of this work is appreciated. However, there are two major revisions during rebuttal:
1. Assumption 1 initially defined $\mathcal{X}$ as a 'countably infinite set' but should be corrected as a 'compact subset spanning $\mathbb{R}^d$'
1. Initial comparison to LINNASH was not fair while corrected numerical results were provided during rebuttal.

Author(s) are highly suggested to adopt reviews to correct and further improve the manuscript.